# Shape control in 2D molecular nanosheets by tuning anisotropic intermolecular interactions and assembly kinetics

Maximilian Dreher [1,3], Pierre Martin Dombrowski [1,3], Matthias Wolfgang Tripp[2], Niels Münster[2], Ulrich Koert[2] & Gregor Witte [1]✉

Since molecular materials often decompose upon exposure to radiation, lithographic patterning techniques established for inorganic materials are usually not applicable for the fabrication of organic nanostructures. Instead, molecular self-organisation must be utilised to achieve bottom-up growth of desired structures. Here, we demonstrate control over the mesoscopic shape of 2D molecular nanosheets without affecting their nanoscopic molecular packing motif, using molecules that do not form lateral covalent bonds. We show that anisotropic attractive Coulomb forces between partially fluorinated pentacenes lead to the growth of distinctly elongated nanosheets and that the direction of elongation differs between nanosheets that were grown and ones that were fabricated by partial desorption of a complete molecular monolayer. Using kinetic Monte Carlo simulations, we show that lateral intermolecular interactions alone are sufficient to rationalise the different kinetics of structure formation during nanosheet growth and desorption, without inclusion of interactions between the molecules and the supporting $MoS_2$ substrate. By comparison of the behaviour of differently fluorinated molecules, experimentally and computationally, we can identify properties of molecules with regard to interactions and molecular packing motifs that are required for an effective utilisation of the observed effect.

Molecular materials are a focus of scientific and technological interest due to the possibility to flexibly tailor their electronic properties. This enables diverse applications ranging from biochemical nanotechnology to organic thin-film electronics[1,2]. In contrast to conventional inorganic electronics that are continuously approaching physical limits as device miniaturisation progresses[3,4], structuring of organic films is still in its infancy. One major reason for this is the radiation sensitivity of organic materials, hampering the application of well-established lithographic techniques for top-down structuring[5,6]. Instead, bottom-up methods are required[7–10], such as molecular self-assembly which provides precise control of molecular packing and scalability for mass fabrication[7,10,11]. This is commonly achieved

through covalent molecule-substrate bonds or covalent intermolecular linkage, often catalysed by metallic substrates, allowing for the fabrication of supramolecular nanostructures such as porous networks or nanoribbons[12–17]. Naturally, such structures are rigid and cannot be reshaped after covalent bonds are formed.

An alternative and so far hardly studied approach is to rely on non-covalent van der Waals (vdW) interactions. Despite their relative weakness, nature has countless examples of complex self-assembly processes, such as DNA association[18] or the organisation of proteinaceous surface layers[19] that rely on non-covalent intermolecular interactions only, proving that such interactions can drive self-assembly of molecules into complex structures. Interestingly,

[1]Department of Physics, Philipps-Universität Marburg, 35037 Marburg, Germany. [2]Department of Chemistry, Philipps-Universität Marburg, 35037 Marburg, Germany. [3]These authors contributed equally: Maximilian Dreher, Pierre Martin Dombrowski. ✉e-mail: gregor.witte@physik.uni-marburg.de

biologic self-assembly processes often require non-equilibrium operation to adopt non-minimum energetic states[20]. Non-equilibrium processes could similarly be used to achieve new means of structural control in artificial molecular nanostructures, for instance through tempering of already assembled vdW structures[21].

In this study, we show that control of the mesoscopic shape of molecular assemblies that preserves molecular packing order can be achieved, which is a step towards true bottom-up patterning through self-assembly. We use regioselective fluorination of parental pentacenes to tune intermolecular interactions and correlate the kinetics of structure formation upon ad- and desorption with these interactions. As support for our molecular nanosheets, we chose molybdenum disulphide ($MoS_2$), which is a popular representative of the class of two-dimensional (2D) materials[22]. $MoS_2$ allows an easy preparation of perfectly flat, long-range ordered surfaces by exfoliation of single crystals[23]. Previous studies have shown that $MoS_2$ exhibits an exceptionally weak molecule-substrate interaction potential, in the absolute strength as well as in its spatial corrugation, for the closely related molecules pentacene and perfluoropentacene[24–27]. As discussed in more detail in the 'Methods' section, this experimental evidence for a negligible corrugation of the molecule-substrate interaction potential shows that structure formation of molecular layers is not influenced by the supporting $MoS_2$ substrate and consequently only determined by intermolecular interactions. Nonetheless, $MoS_2$ is sufficiently conductive for scanning tunnelling microscopy (STM) measurements, in contrast to similarly inert substrates such as hexagonal boron nitride.

## Results

### Growth and desorption of L-F₆PEN nanosheets

The first molecule chosen for our study is 1,2,10,11,12,14-hexafluoropentacene (L-$F_6$PEN, $C_{22}H_8F_6$)[28], a pentacene derivative that is fluorinated unilaterally adjacent the molecular L-axis (cf. Fig. 1b). Due to the electronegativity of fluorine, its outer electrostatic potential, computed using effective charges for all atoms obtained from a natural bond orbital analysis (cf. 'Methods'), is bipolar with a nodal line along

the long molecular L-axis (cf. Fig. 1b). Figure 1a shows STM data of a nominal monolayer of L-$F_6$PEN grown on an exfoliated $MoS_2$(001) crystal, prepared by deposition of a nominal film thickness of 3.0 Å as measured by a quartz crystals microbalance. The molecules adsorb in a coplanar orientation on the substrate surface as confirmed by X-ray absorption measurements (cf. Supplementary Note 1) and form extraordinarily large, single-crystalline domains extending over 200 nm (cf. Supplementary Note 2). From the calibrated STM data, we find an oblique unit cell with a = 16.6 ± 1.0 Å, b = 7.3 ± 1.0 Å and $\gamma = 70° \pm 5°$ that can be rationalised as a $\begin{pmatrix} 1.64 & 1.00 \\ 5.76 & -4.50 \end{pmatrix}$ point-on-line superstructure on $MoS_2$ (cf. blue unit cell in Fig. 1c). Note that, due to the 6-fold rotational symmetry of the substrate surface and the presence of mirror domains, the molecular adlayers occur in 12 azimuthal orientations. This structure is in good agreement with an energetically optimised 2D structure based on the interactions discussed in the 'Methods' section (a = 16.78 Å, b = 7.33 Å, $\gamma$ = 70.81°), confirming the negligible influence of the corrugation of the molecule-substrate interaction potential on the adlayer structure. Submonolayer films with a nominal thickness of 0.8 Å, grown by deposition of material onto the substrate surface, form well-ordered islands (cf. Fig. 1d). These islands adopt the same unit cell as the saturated monolayer and exhibit a characteristic, elongated shape in direction $\langle \vec{b} \rangle$ as illustrated in Fig. 1f. This preferred direction of elongation was also observed in films with different surface coverages, as discussed in Supplementary Note 3. By contrast, annealing of a complete monolayer at 400 K for 1 min to activate partial desorption of molecules yields submonolayer islands with an elongated shape in direction $\langle \vec{a} \rangle$ (cf. unit cell in Fig. 1c), as shown in Fig. 1e and illustrated in Fig. 1g. Between the elongated islands, there are some tiny, roundish islands consisting of 20–30 molecules that are too small to show a clear shape preference. Previous temperature-programmed desorption (TPD) experiments of pentacene and perfluorinated pentacene films adsorbed on $MoS_2$(001) showed notable desorption of the first molecular monolayer at temperatures above 400 K[25]. For the similar L-$F_6$PEN, we have tested

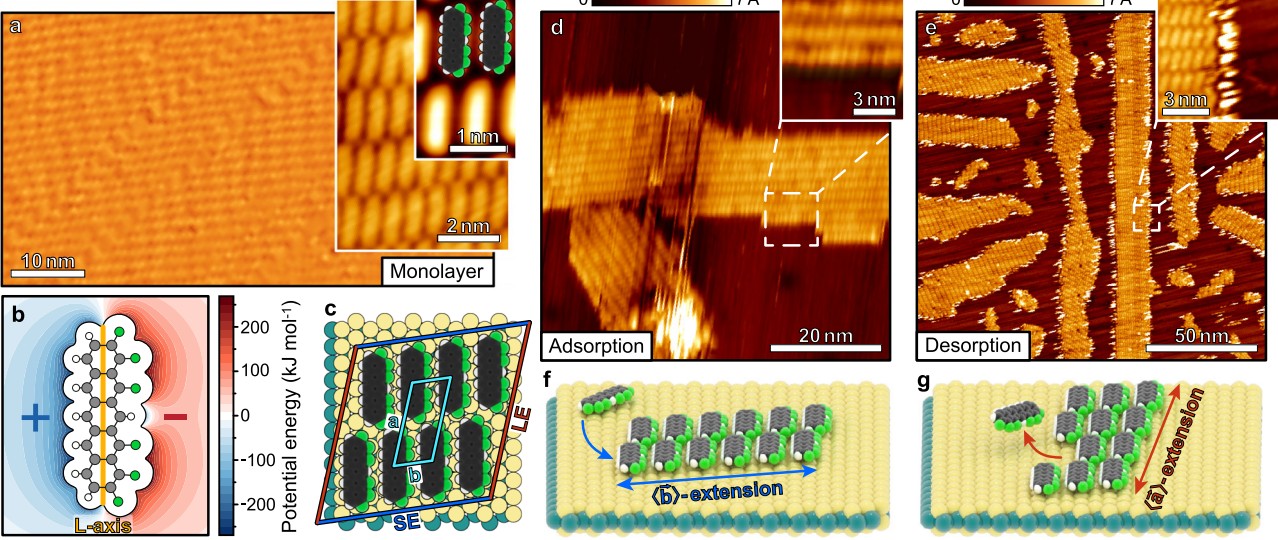

**Fig. 1 | Structure of L-F6PEN nanosheets. a** STM micrograph of a complete monolayer of L-$F_6$PEN adsorbed on $MoS_2$ (U = 1.2 V, I = 40 pA). The large scan shows a smooth and closed film, the insets show the molecular packing motif. **b** Plot of the outer electrostatic potential of an L-$F_6$PEN molecule ($C_{22}H_8F_6$) probed by an electron (cf. 'Methods'). Negative values are blue, positive ones red. The white area in the centre corresponds to the vdW box of the molecule. Carbon atoms are shown in grey, hydrogen atoms in white, and fluorine atoms in green. The orange line marks the L-axis of the molecule. **c** Illustration of the unit cell of the L-$F_6$PEN (sub-)

monolayer (light blue rhombus) together with denotations for the island edges. The long side edges (LEs) are marked red, the short side edges (SEs) in dark blue. The depicted 2 × 4 nanosheet was chosen arbitrarily and bears no specific meaning. **d, e** STM micrographs of islands of L-$F_6$PEN adsorbed on $MoS_2$ (top) created by direct deposition (nominal thickness 0.8 Å; U = 1.2 V, I = 40 pA) and thermal treatment after deposition of a monolayer (400 K for 1 min; U = 1.2 V, I = 50 pA), respectively. **f, g** Illustrations of the island shapes observed upon adsorption and desorption, respectively.

different annealing temperatures between 390 K and 420 K, finding no deviation from the preferred nanosheet elongation in direction $\langle \vec{a} \rangle$ from the very onset of desorption (cf. Supplementary Note 4) to the complete desorption of the initial monolayer as validated by STM prior to and after annealing. Considerations about the boundary free energy (the 2D analogue of the surface free energy, cf. Supplementary Note 5) suggest that the $\langle \vec{b} \rangle$-extended sheets are much closer to the energy local minimum. Hence, it appears counterintuitive, that $\langle \vec{b} \rangle$-extended sheets are found experimentally prior, but not after thermal treatment. The distinctly different preferential nanosheet extensions in directions $\langle \vec{a} \rangle$ and $\langle \vec{b} \rangle$ occurring upon growth and desorption, respectively, show that the kinetics of structure formation provide means to control the shape of such molecular islands while retaining their crystalline molecular packing arrangement.

To rationalise these differences between ad- and desorption and their influence on the resulting sheet shape, we have simulated both processes using Monte Carlo approaches. Intermolecular interactions were modelled following the work by Kröger et al.[29], assigning effective interaction parameters for Coulomb, Pauli, and dispersion interactions to each atom of a molecule (cf. 'Methods'). To distinguish between the different edges of a nanosheet for attachment and desorption processes, we denote them according to the molecular side that is exposed at the respective nanosheet edge as illustrated in Fig. 1c: At the red-coloured nanosheet edges, the long sides of molecules are exposed, hence they are denoted as long side edges (LEs). At the blue edges, the short sides of the molecules are exposed, therefore they are denoted as short side edges (SEs). Figure 2a shows an attachment energy map (AEM), i.e., the potential energy of a probe molecule in the vicinity of a fixed molecule as a function of their relative centre-to-centre displacement ($\Delta x$, $\Delta y$) for two equally oriented L-F$_6$PEN molecules. Here, blue and red areas correspond to attractive and repulsive interactions, respectively. The white area in the centre of the AEM, sometimes referred to as excluded area (cf. inset Fig. 2a, b), marks the area that is physically inaccessible due to Pauli repulsion preventing mutual overlap of the laterally interacting molecules. Along the LEs, there is a strong attraction caused predominantly by Coulomb interactions, whereas we find repulsive interactions along the SEs. Consequently, for a system consisting of only two molecules, attachment at the LEs is energetically favoured. This tendency is also found for islands of L-F$_6$PEN molecules as shown by the AEM in Fig. 2b, where we see again strongest attractions along the LEs. In contrast to the single-

molecule AEM, there are small cusps of weak attraction along the SEs that promote continuation of the crystal lattice also at the SE (cf. Supplementary Fig. 7). However, to reach these small minima of the AEM, a potential barrier must be overcome kinetically. A more detailed analysis considering different azimuthal orientations of the probe molecules is given in Supplementary Note 6.

To validate that this anisotropy of the attachment energy favours attachment of admolecules at the LEs and hence the formation of $\langle \vec{b} \rangle$-extended nanosheets, we must also consider the kinetics of growth. Therefore, we have simulated the initial attachment process of single molecules to a nanosheet with close-packed edges and determined the probability of attachment at the LEs, $p$(LE), in dependence of the 2D sheet dimensions (cf. 'Methods'). The results are shown in Fig. 2c, where the colour of the element in the $m_a^{th}$ row and the $n_b^{th}$ column, counted from the bottom left corner, corresponds to $p$(LE) for an island of $m_a \times n_b$ molecules, as exemplarily illustrated in the left panel for a $3 \times 2$ island. As one would expect, the geometrical aspect ratio of the nanosheet has a notable influence on $p$(LE), which tends to be larger for $\langle \vec{a} \rangle$-extended nanosheets (area above the yellow line in Fig. 2c) than for $\langle \vec{b} \rangle$-extended sheets (area below yellow line). Nonetheless, with the exception of small clusters consisting of few molecules or nanosheets with an extreme extension in direction $\langle \vec{b} \rangle$ (such as a single row of molecules in that direction), $p$(LE) is always larger than 50%. Therefore, attachment of additional molecules to a nanosheet always favours $\langle \vec{b} \rangle$-extended growth, as $p$(LE) mostly exceeds 50%, and any growth of a sheet in direction $\langle \vec{a} \rangle$ leads to an increase of $p$(LE), thus further promoting growth in direction $\langle \vec{b} \rangle$.

In addition to the kinetics of growth, we have also simulated the kinetics of desorption (cf. 'Methods'). Supplementary Movie 1 shows the desorption of a single L-F$_6$PEN island consisting of $m_a = 20$ by $n_b = 50$ molecules. A map of the average desorption sequence of molecules from this nanosheet, determined from 50 individual simulations of desorption of such a nanosheet, is shown in Fig. 2d. Though the initial island shape has no pronounced geometrical extension in either $\langle \vec{a} \rangle$ or $\langle \vec{b} \rangle$ direction, a shape that is extended in $\langle \vec{a} \rangle$ direction is clearly preferred after partial desorption of molecules, in agreement with our experimental observations. Simulations with different initial nanosheet shapes (cf. Supplementary Note 7) yield the same results, thereby showing that the preference of $\langle \vec{a} \rangle$-extension is independent of the initial island shape. Considerations of the cohesive energy of individual molecules in a nanosheet (cf. Supplementary Note 8) show that molecules located at the LEs of a nanosheet have the weakest bond with the nanosheet. While molecules located at the SEs have two

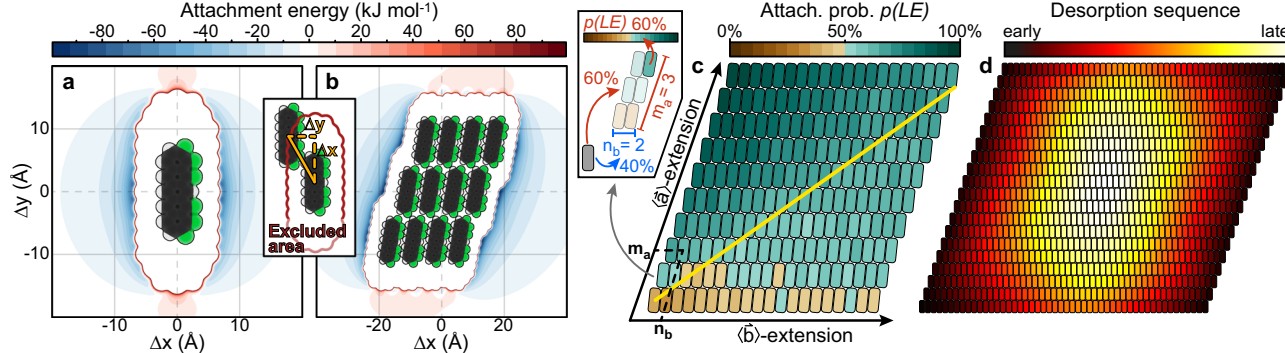

**Fig. 2 | Computational analysis of L-F6PEN nanosheet formation.**
**a**, **b** Attachment energy maps (AEMs) for a single L-F$_6$PEN molecule and a $3 \times 4$ island of L-F$_6$PEN molecules, respectively, probed by an equally oriented L-F$_6$PEN molecule. Inset: Illustration of the boundary of the excluded area (red) that marks relative displacements that are physically inaccessible due to Pauli repulsion.
**c** Probability $p$(LE) for attachment of an L-F$_6$PEN admolecule to a long side edge (LE) for a given island shape of $m_a$ molecules in direction $\langle \vec{a} \rangle$ and $n_b$ molecules in

direction $\langle \vec{b} \rangle$, as function of the island shape $m_a \times n_b$. Note that $p$(LE) = 1 − $p$(SE). The yellow line marks a geometrical aspect ratio of the nanosheets of 1:1. Inset: Illustrated reading instruction for the map. The colour of the molecule in the top right corner of the exemplary $3 \times 2$ island denotes $p$(LE) for this island shape, in this case 60%. **d** Map of the desorption sequence of L-F$_6$PEN molecules from a $20 \times 50$ island averaged from 50 simulated temperature-programmed desorption experiments.

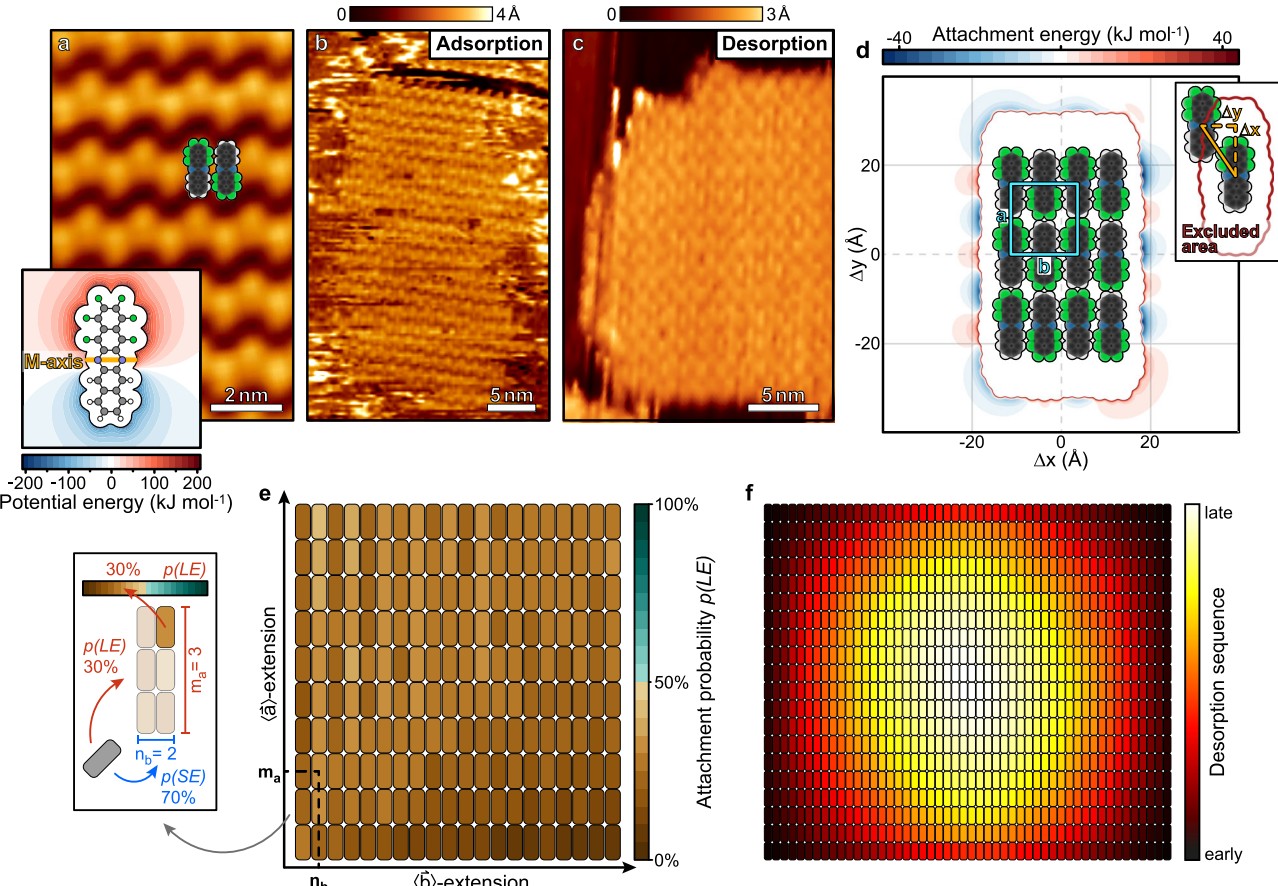

**Fig. 3 | Experimental and computational analysis of M-F$_6$PEN nanosheet formation. a** STM micrograph of a complete monolayer of M-F$_6$PEN adsorbed on MoS$_2$ (U = 2.0 V, I = 400 pA). Inset: Plot of the outer electrostatic potential of a M-F$_6$PEN molecule (C$_{20}$N$_2$H$_6$F$_6$) probed by an electron. Carbon atoms are shown in grey, hydrogen atoms in white, fluorine atoms in green, and nitrogen atoms in blue. The orange line marks the M-axis of the molecule. **b**, **c** STM micrographs of islands of M-F$_6$PEN adsorbed on MoS$_2$ created by direct deposition (nominal thickness 0.8 Å; U = 2.0 V, I = 450 pA) and thermal treatment after deposition of a monolayer (410 K for 1 min, U = 1.3 V, I = 50 pA), respectively. **d** Attachment energy map (AEM) of a 3 × 4 island of M-F$_6$PEN molecules. The blue rectangle marks the unit cell of the

molecular lattice. Inset: Illustration of the orientation of the probe molecule used for computation of the AEM and illustration of the boundary of the excluded area (red) that marks relative displacements that are physically inaccessible due to Pauli repulsion. **e** Probability $p$(LE) for attachment of an M-F$_6$PEN admolecule at the long side edge (LE) for a given island shape. Inset: Illustrated reading instruction for the map. The colour of the molecule in the top right corner of the exemplary 3 × 2 island denotes $p$(LE) for this island shape, in this case 30%. **f** Map of the desorption sequence of M-F$_6$PEN molecules from a 20 × 50 island averaged from 50 simulated temperature-programmed desorption experiments.

long-side nearest neighbours (with a stronger mutual attraction between molecules), those at the LEs have only one long-side neighbour and therefore significantly more likely to desorb than molecules located at the SEs. Thus, the same interaction that promotes $\langle \vec{b} \rangle$-extended growth of nanosheets leads to the formation of $\langle \vec{a} \rangle$-extended nanosheets upon partial desorption.

**Growth and desorption of M-F$_6$PEN nanosheets**

To further validate our findings and examine the influence of the fluorination pattern of the molecule on the shape of nanosheets, we also analysed the nanosheet formation of another partially fluorinated pentacene derivative, 1,2,3,4,5,14-hexafluoro-6,13-diazapentacene (M-F$_6$PEN, C$_{20}$N$_2$H$_6$F$_6$)[30]. This molecule is fluorinated above the median molecular M-axis and exhibits a distinctly bipolar electrostatic potential along the L-axis (cf. inset Fig. 3a). Figure 3a depicts a STM micrograph of a complete monolayer of M-F$_6$PEN on MoS$_2$. A distinct contrast between the fluorinated and non-fluorinated halves of the molecule is visible. Interestingly, the azimuthal orientation of the molecules is alternating in direction $\langle \vec{b} \rangle$, in contrast to the uniformly oriented L-F$_6$PEN. Figure 3b shows an STM micrograph of a sub-monolayer (d$_{nom}$ = 0.8 Å) M-F$_6$PEN on MoS$_2$. Here, as for adsorption, molecular islands are preferentially elongated in direction $\langle \vec{a} \rangle$,

opposite to L-F$_6$PEN nanosheets. By contrast, heating of a complete monolayer of M-F$_6$PEN to 410 K for 1 min to induce partial desorption yields islands with no clear shape preference (cf. Fig. 3c), unlike in the L-F$_6$PEN case.

As for L-F$_6$PEN, we performed simulations to understand this behaviour. Figure 3d shows an AEM of a 3 × 4 sheet of M-F$_6$PEN. Due to the alternating orientation of molecules, the interaction range of the M-F$_6$PEN nanosheet is much shorter than that of a corresponding L-F$_6$PEN nanosheet (cf. 2b). Though strongest attraction is found along the LEs, there are weaker but longer-ranged attractive interactions along the SEs. The longer range of those attractive interaction might promote attachment at the SEs despite overall weaker attraction. Indeed, in agreement with the experimental results, Monte Carlo simulations of the single-molecule attachment process to nanosheets (cf. Fig. 3e) show a clear preference for the formation of $\langle \vec{a} \rangle$-extended nanosheets, as $p$(LE) is always below 50% and quite homogeneous. In a simple picture, the alternating attractive and repulsive cusps at the LEs (cf. Fig. 3d) cancel all electrostatic interactions in the far field, but not in the near field. In contrast, the L-F$_6$PEN islands exhibit also a long-range repulsive nature in the far field at the SEs (cf. Fig. 2b). Consequently, $p$(LE) is almost independent of the nanosheet shape in the case of M-F$_6$PEN, but shows a distinct dependence on the nanosheet

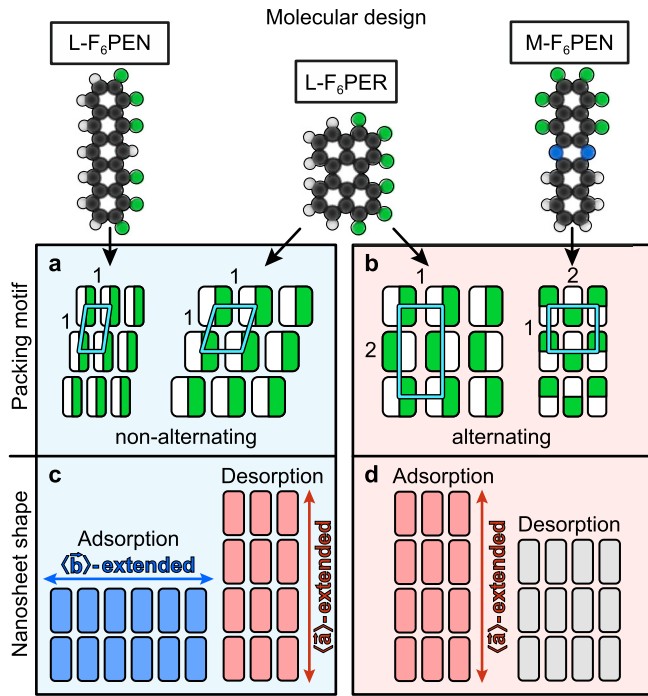

**Fig. 4 | Summary of the observed nanosheet shapes and packing motifs.**
**a, b** Stable packing motifs of L-F$_6$PEN, L-F$_6$PER, and M-F$_6$PEN with unit cells marked in light blue. In the illustrations of the packing motifs, the predominantly hydrogen-terminated sides of molecules are coloured white and the predominantly fluorine terminated halves are coloured green. The carbon backbones (grey) and nitrogen atoms (blue) are not shown in the illustrations of packing motifs. Numbers indicate the numbers of molecules in the unit cell along the corresponding unit cell edge. **c, d** Shape of nanosheets that are formed upon growth and partial desorption for these packing motifs. $\langle b \rangle$-extended nanosheets are coloured blue, $\langle a \rangle$-extended sheets are coloured red and sheets with no distinct extension are coloured grey.

shape for the L-F$_6$PEN. Simulations of the dismantling process during desorption of an M-F$_6$PEN nanosheet (Fig. 3f) show no preferred extension, in agreement with our experimental results. The reason for the absence of a preferred edge for desorption can again be found in the cohesive energies of molecules in an M-F$_6$PEN nanosheet (cf. Supplementary Note 8): The absolute difference between the cohesive energies of molecules located at the LEs and SEs is significantly smaller than for L-F$_6$PEN, so that desorption probabilities are almost equal at all edges.

## Discussion

Our results show that anisotropic intermolecular interactions can result in markedly different island shapes for sheets that are grown vs. ones that were created by partial desorption from complete molecular monolayers. Thus, in the case of L-F$_6$PEN, it is possible to selectively fabricate islands that are elongated either in direction $\langle a \rangle$ or $\langle b \rangle$ via growth or partial desorption, respectively. For M-F$_6$PEN, growth yields nanosheets that are extended in direction $\langle a \rangle$, while partial desorption yields nanosheets without a distinct elongation in either direction. Though these results do serve as a proof of concept for the control of nanosheet shapes via the variation of the preparation protocol, further analysis is required to understand the reasons for different responses of the two molecules to ad- and desorption processes, and to allow the targeted design of molecules for specific responses to the respective processes.

Since structural differences in nanostructures are often attributed to kinetically or energetically controlled growth, we have also analysed the shape dependence of the nanosheet energy. Considerations of the boundary free energy of our nanosheets (cf.

Supplementary Note 5), the 2D equivalent of the surface free energy of crystals, suggest that $\langle b \rangle$-extension is energetically preferred for L-F$_6$PEN and M-F$_6$PEN. However, even the LE-extended sheets observed here deviate distinctly from the energetically optimal aspect ratio. This shows that both analysed formation processes are kinetically controlled and do not yield an energetically optimised sheet shape, reflecting the fact that vacuum-based growth is not an equilibrium process.

To examine the influence of the molecular shape on the growth kinetics upon ad- and desorption, we have extended our computational analysis to the partially fluorinated perylene derivative 1,2,3,10,11,12-hexafluoroperylene (L-F$_6$PER, C$_{20}$H$_6$F$_6$; cf. top panel of Fig. 4). Like L-F$_6$PEN, this molecule is fluorinated along one long side. However, it is not as elongated as the pentacene derivatives, but has a more compact, almost square shape. Although this molecule has not been synthesised yet and no experimental data is available, we use our computational model to provide predictions (cf. Supplementary Note 9). Due to the more compact shape of L-F$_6$PER in comparison to the pentacene derivatives, a uniform molecular orientation as observed for L-F$_6$PEN energetically is not clearly favoured over an alternating molecular orientation like the structure observed for M-F$_6$PEN (cf. Fig. 4). Though the alternating structure is energetically slightly more stable by 8%, it is likely that both structures can occur for L-F$_6$PER under the right preparation conditions. Therefore, we have simulated ad- and desorption processes for both packing motifs. Interestingly, the shape of the molecular islands depends decisively on the molecular packing motif, as illustrated in the bottom panel of Fig. 4: In case of a uniform molecular orientation as observed for L-F$_6$PEN, island shapes formed upon ad- and partial desorption closely resemble those observed for L-F$_6$PEN (cf. Supplementary Note 9). In the case of an alternating molecular orientation, ad- and desorption kinetics closely resemble those observed for M-F$_6$PEN. This suggests that the different growth and desorption behaviours are caused by the molecular packing motif, as illustrated in Fig. 4: For the packing motifs with uniform molecular orientation, nanosheets have two edges that are terminated with hydrogen or fluorine only, which are the LEs for L-F$_6$PEN and L-F$_6$PER. Along the uniformly terminated LEs, the inter-action potential for equally oriented attaching molecules is strongest and purely attractive, hence promoting attachment of molecules. In this case the molecules, at first glance somewhat paradoxically (but analogously to detachments from L-F$_6$PEN nanosheets), have the weakest bond to the nanosheet and therefore tend to desorb first, leading to the formation of islands elongated in the $\langle a \rangle$ direction. For the packing motifs with alternating molecular orientations, kinetics are more complex due to the less homogeneous interaction potential of nanosheets that feature alternating regions of attraction and repulsion along all edges, which also reduces the range of interactions (cf. Supplementary Note 6). For both alternating packing motifs studied in this work, we find a preferential direction for nanosheet extension upon adsorption, but no such preference upon partial desorption.

These findings allow us to formulate criteria for molecules that are prerequisite for the fabrication of anisotropically shaped nanosheets that enable a structural control through ad- and desorption: (i) Molecules must interact attractively to promote the formation of stable nanosheets. (ii) Interactions must be anisotropic, which also requires shape anisotropy of the molecule. If interactions are isotropically attractive, no preferential direction for lateral bonding of molecules can be observed. (iii) Molecules must prefer a packing motif that allows the formation of edges with a homogeneous termination. These edges are crucial for opposite nanosheet extensions upon growth and partial desorption, as they are both the preferred edges for ad- and desorption as discussed above. (iv) Interactions must be sufficiently strong to prohibit detachment of molecules from the nanosheet without desorption. If molecules could detach from nanosheets during thermal treatment and reattach elsewhere, it is likely that no shape difference

between nanosheets that were grown and ones that were created by partial desorption would occur.

In summary, we rationalise the kinetically controlled formation of 2D molecular nanosheets composed of partially fluorinated polycyclic aromatic hydrocarbons with bipolar electrostatic potentials using high-resolution STM data and simulations of ad- and desorption processes. We find that packing motifs that allow the formation of uniformly, i.e., only hydrogen- or fluorine-terminated edges, promote attachment of molecules to and detachment of molecules from these edges upon ad- and desorption, respectively. This provides means of control over the shape of nanosheets that are elongated in one direction upon growth of the sheets and in the other upon desorption. A key property of the molecules used in this work is the pattern of fluorination and their shape anisotropy. In case of distinctly elongated molecules with partial fluorination along one long side only (like for L-F$_6$PEN), a packing motif with uniform molecular orientation will occur, allowing the formation of uniformly terminated nanosheet edges.

Since the mechanism of nanosheet shape control identified here is essentially based on the Keesom forces of the intermolecular van der Waals interactions, it is hardly applicable to molecular films adsorbed on metal substrates, where additional molecule-substrate interactions occur that affect the molecular thin film structure and that typically exceed lateral intermolecular interactions in strength. However, given the large class of 2D materials (such as transition metal dichalcogenides, hexagonal boron nitride, graphene, or others), this is not a material niche. In particular with regard to the interesting potential of organic/2D material hybrid systems for future device architectures[31,32], in which a lateral structuring of molecular adlayers is hardly possible, the method for shape control described in this work can offer an interesting approach to the production of molecular nanosheets with defined shapes.

## Methods

### Experimental

L-F$_6$PEN and M-F$_6$PEN films were grown by means of organic molecular beam deposition under ultrahigh vacuum (UHV) conditions from resistively heated Knudsen cells at room temperature with typical deposition rates of 2 Å/min as determined by quartz crystal microbalances. Molecular materials with an initial purity of approximately 98% were carefully degassed before deposition to increase material purity. MoS$_2$ crystals were grown by means of chemical vapour transport from stoichiometric amounts of Mo and S using Br$_2$ as source for the transport agent MoBr$_4$ formed in situ. The reaction was performed in an evacuated quartz glass ampoule with a temperature gradient from 1300 K to 1220 K for 20 days[23]. The MoS$_2$ crystals were exfoliated under ambient conditions prior to evacuation and annealed at 650 K for 15 min prior to the deposition of organic material. STM measurements were carried out in UHV (base pressure <10$^{-10}$ mbar) using an Omicron VT STM in constant current mode with etched tungsten tips at sample temperatures of 110 K. Complementary work function measurements were conducted in situ in the same UHV system by means of a Kelvin probe setup (Besocke Delta Phi GmbH, Kelvin Probe S). As a reference for the contact potential, we used highly oriented pyrolytic graphite (HOPG, NT-MDT, quality: A) with a work function of 4.4 eV[33]. NEXAFS measurements in partial electron yield mode (retarding field: 150 V) were performed at the HE-SGM beamline of the electron storage ring BESSY II in Berlin (Germany). Details on the experimental setup and the data analysis can be found in ref. [34].

### Computational

Attachment and desorption processes were simulated using self-programmed code. The source code is available free of charge (cf. 'Code availability') with a detailed documentation. Here, we briefly describe the underlying model and assumptions for the simulations performed throughout this work.

One fundamental assumption underlying all molecular dynamics simulations is that the corrugation of the molecule-substrate interaction potential is negligible. This assumption might appear crude first, but it is based on experimental evidence reported for pentacene (C$_{22}$H$_{14}$, PEN) and perfluoropentacene (C$_{22}$F$_{14}$, PFP), two closely related molecules to the L-F$_6$PEN and M-F$_6$PEN used in this work. In ref. [25], it is shown by means of temperature-programmed desorption (TPD) and STM that the repulsively interacting acenes are highly mobile on the MoS$_2$(001) surface, even at cryogenic temperatures at which well-ordered films of the same molecules are observable on metal surfaces. In ref. [24], a point-on-line epitaxy is reported for multilayer films of PFP grown on different transition metal dichalcogenides, including MoS$_2$. The observed structure shows that growth of the molecular film is indeed not dictated by molecule-substrate interactions in the first layer of the film, but only by intermolecular interactions. Finally, ref. [27] shows that PEN deposited on 2D flakes of MoS$_2$ on a Au(111) substrate adsorbs preferentially on the Au(111) substrate, showing that molecules can diffuse from the MoS$_2$ flake to the Au(111) substrate and that the adsorption energy for PEN on Au(111) is significantly larger than that for PEN on MoS$_2$, which is supported by DFT computations and experimental data in refs. [25,35]. Together, these studies show that the molecule-substrate interaction potential and its spatial corrugation must be negligibly small, as it appears that all structure formation is purely dictated by intermolecular interactions. For these reasons, we have neglected any corrugation of the molecule-substrate interaction potential and represent the molecule-substrate interaction through a constant term that is only relevant for TPD simulations. Such simulations with the same assumption were already performed for PEN and PFP adsorbed on MoS$_2$ in ref. [25], successfully replicating experimental data.

Since L-F$_6$PEN and M-F$_6$PEN are both planar molecules, adsorb coplanar on the MoS$_2$(001) surface, and grow in a single layer, molecular diffusion is restricted to two dimensions. Intermolecular interactions were modelled as described by Kröger et al.[29], considering electrostatic interactions between the intramolecular charge distributions, attractive dispersion forces, and short-ranged Pauli repulsion. In absence of any interface dipole moments, as supported by work function measurements (cf. Supplementary Note 10) and other studies[25,26], no further interactions have to be taken into account. Though intermolecular interactions of molecules with C-H groups and highly electronegative elements are sometimes described as hydrogen bonds[15], we want to emphasise that these are not relevant here. H-F distances in the observed crystalline lattices are predominantly around 2.4 Å, and all above 2 Å, at which hydrogen bonds fall in the weak and purely electrostatic category[36]. Hence, our model of electrostatic interactions includes what some might consider hydrogen bonds.

Since we need to appropriately model electrostatic interactions in the near field, we cannot rely on simple far-field approximations of dipole-dipole interactions but need a finer subdivision of the intramolecular charge distributions. Therefore, all intermolecular interactions were treated in an atomistic model, in which molecules are modelled as rigid structures consisting of, in the cases of L-F$_6$PEN, M-F$_6$PEN, and L-F$_6$PER, 36, 34, and 32 atoms, respectively, that each have specific interaction parameters. Dispersion interactions and Pauli repulsion are modelled together in a Buckingham potential: $V_B = a_{ij} \exp(-b_{ij} r_{ij}) - c_{ij}/r_{ij}^6$, where $r_{ij}$ is the distance of the two interacting atoms $i$ and $j$ and $a_{ij}$, $b_{ij}$, and $c_{ij}$ are element specific material parameters. Interaction parameters for symmetric atomic pairs, i.e., $i = j$ for carbon, hydrogen, and nitrogen were taken from ref. [29]. Parameters for fluorine-fluorine interactions stem from ref. [37]. For asymmetric interactions, i.e., $i \neq j$, interaction parameters were approximated by the geometric mean of the interaction parameters of the involved elements: $a_{ij} = \sqrt{a_{ii} a_{jj}}$, $b_{ij}$ and $c_{ij}$ accordingly.

Electrostatic interactions were modelled by attributing charges to each atom that interact via Coulomb interaction. These charges were determined from natural bond orbital (NBO) analyses using NBO 7.0[38]. The NBO analyses are based on atomic basis sets from DFT calculations performed using GAMESS (version 2020.R2) with the B3LYP functional and aug-CC-pvTZ basis set[39]. All interaction parameters used in this work can be found in Supplementary Note 11.

## Interaction potential maps

Based on the above-described intermolecular interactions, interaction potential maps were calculated by positioning a single molecule or a rigid molecular nanosheet with fixed orientation and location in the centre of the coordinate system and then calculating the interaction energy of a probe particle of fixed orientation with the centre molecule or island as a function of the relative probe displacement. For calculations of the outer molecular electrostatic potential (Fig. 1a and inset of Fig. 4a), an electron was used as a probe particle. For attachment energy maps (AEMs), molecules were used as probes. High-resolution AEMs were used to optimise the crystal structure determined by STM for subsequent simulations. These optimised geometries are in good agreement with the experimental STM data. For L-$F_6$PEN, optimisation yields a = 16.78 Å, b = 7.33 Å, and γ = 70.81° with an angle of 80° between the molecular L-axis and b. For M-$F_6$PEN, we use a = 16.4 Å, b = 15.0 Å, and γ = 90° with an angle of 90° between the molecular L-axis and b and two molecules per unit cell. For the uniformly oriented L-$F_6$PER structure, we find a = 12.06 Å, b = 9.14 Å, and γ = 65.6° with an angle of 84.4° between the molecular L-axis and b. For the alternating L-$F_6$PER structure, we find a unit cell with a = 23.58 Å, b = 9.24 Å, and γ = 90° with an angle of 90° between the molecular L-axis and b and two molecules per unit cell.

## Attachment at nanosheets

Molecular attachment at existing nanosheets was modelled by creation of a rigid molecular nanosheet consisting of $m_a \times n_b$ molecules. A single additional molecule was placed at a random location on a ring around the nanosheet centre with a radius of thrice the width or height of the nanosheet, whichever is larger. The admolecule was then allowed to move for a fixed number of 100 steps along the force vector acting at its current location. The step size is coupled to the admolecule-nanosheet distance so that it decreases as the molecule approaches the nanosheet. After completion of the movement steps, the location of the admolecule is categorised as follows: If its distance to the closest molecule exceeds 1.2 times the diagonal of the unit cell of the structure, it is considered to not have attached. Else, the closest edge is determined to determine whether the admolecule has attached to an LE or an SE. This simulation was performed 1000 times per nanosheet size ($m_a \times n_b$) and repeated for varying nanosheet sizes, allowing for the calculation of an LE or SE attachment probability, $p$(LE) or $p$(SE) = 1 − $p$(LE), respectively, as a function of the nanosheet LE and SE extension.

## Desorption from nanosheets

To analyse desorption of molecules from nanosheets, TPD was simulated based on an algorithm described by Meng and Weinberg in ref. [40]. This algorithm was already successfully used to simulate TPD of PEN and PFP from MoS$_2$ substrates[25]. Therefore, a surface is populated with $N$ molecules in a nanosheet according to the optimised unit cell. For the partially fluorinated species, rigid nanosheets were assumed, i.e., molecules were not allowed to diffuse on the surface. This assumption is founded on the experimental observation that nanosheets created by partial desorption from complete layers do not adopt the energetically optimised ⟨$\bar{b}$⟩-extended shape (cf. Fig. 3 and Supplementary Note 5), which would require temporary detachment from the island and reattachment at other sites, thus allowing us to exclude such processes. This is also unlikely for energetical reasons: Since the strong

lateral intermolecular interactions can exceed the molecule-substrate binding energy, it is likely that molecules would desorb immediately after detachment at the temperatures required to activate detachment from a nanosheet due to the sudden drop of the activation energy of desorption by ~50%.

Desorption was simulated based on the reaction rate constant $k(T) = \nu \exp[-(E_0 - V_{int})/(RT)]$, where $\nu$ is the prefactor of desorption, $E_0$ is the activation energy of desorption in the limit of an isolated molecule on the surface, $V_{int}$ is the interaction energy of the molecule for which $k(T)$ is calculated with its surroundings, $R$ is the universal gas constant, and $T$ is the surface temperature. $\nu$ and $E_0$ define the desorption kinetics, as explained in detail in refs. [25,35]. The prefactor $\nu$, depends predominantly on the surface dynamics. Since we expect a rigid, crystalline structure for the nanosheet even at elevated temperatures, we have chosen the closest available experimental value of $10^{19}$ s$^{-1}$ for a crystalline monolayer of PEN[35]. $E_0$ is directly related to the molecule-substrate binding energy. This parameter has already been determined for two related molecules, PEN and PFP, adsorbed on the MoS$_2$(001) surface. Since optoelectronic properties such as energy levels and frontier orbitals of partially fluorinated pentacenes usually lie between those of PEN and PFP[34,41], and $E_0$ is almost equal for PEN adsorbed on MoS$_2$ and PFP adsorbed on MoS$_2$ with 122 ± 5 kJ mol$^{-1}$ and 131 ± 4 kJ mol$^{-1}$, respectively, we have chosen the experimental data for PEN of 122 kJ mol$^{-1}$. Note that, since not quantitative conclusions are drawn from the TPD simulations, the exact values for $\nu$ and $E_0$ are not critical. In absence of any available experimental data, the same parameters were used for TPD simulations of L-$F_6$PER.

The TPD algorithm proceeds as follows: The reaction desorption rate constant $k(T)$ is calculated for all molecules individually according to the interaction potential with their surroundings. To reduce computational workload, intermolecular interactions were limited to a radius of 50 Å around each molecule. Inclusion of interactions between all molecules changes the interaction energy of a molecule by less than 1% and can therefore be neglected. After calculating all reaction rate constants, a desorption probability is defined as $p_i = k_i/k_{max}$, where $k_i$ is the reaction rate constant for molecule $i$ and $k_{max}$ is the current maximum reaction rate constant of all adsorbed molecules. A molecule is randomly chosen and allowed to desorb with the probability $p_i$. If it does not desorb, the random choice is repeated until desorption occurs. Upon desorption, the molecule is removed from the surface, the surface temperature is increased by $\beta/\sum_i k_i$, assuming a heating rate of $\beta = 1$K/s for the simulated TPD experiments, and all reaction rates are updated. This process is repeated until no molecules remain on the surface. TPD experiments were simulated for various initial nanosheet shapes. In any case, at least 1000 molecules were used per simulated experiment, and simulations were conducted 50 times per nanosheet shape to obtain reliable statistical information.

For validation, TPD simulations were also carried out for PEN. Here, due to repulsive intermolecular interactions, molecular diffusion must be included[25]. Therefore, between individual desorption events, molecules were allowed to diffuse for a predetermined time period. Initially, all molecules were given Boltzmann-distributed random initial (angular-) velocities that were increased according to the temperature increase between desorption events. Periodic boundary conditions were included to allow for smooth diffusion in the confined sample surface area. Due to high computational demand of diffusion simulations that is further increased by the periodic boundary conditions, diffusion time was limited to 1 ps at low coverage and reduced proportional to the surface coverage at higher coverages. TPD of PEN/MoS$_2$ was simulated for varying initial surface coverages (cf. Supplementary Note 12) and is in reasonable agreement with the experimental data published in ref. [25], thus proving that the interaction potential used in this work are reasonably accurate.

## Reporting summary

Further information on research design is available in the Nature Portfolio Reporting Summary linked to this article.

## Data availability

Source data are provided as a repository under the following URL: https://zenodo.org/record/7674282[42]. Any additional data is available from the authors upon request.

## Code availability

The source code required to replicate the computational results presented in this study is provided as a repository under the following URL: https://zenodo.org/record/7319535[43].

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

## Acknowledgements

We gratefully acknowledge financial support from the German Research Foundation (DFG) in the framework of the Collaborative Research Centre 'Structure and Dynamics of Internal Interfaces' (grant no. 223848855-SFB 1083, TP A2 and TP A8; support received by all authors) as well as the Helmholtz-Zentrum Berlin (electron storage ring BESSY II) for provision of synchrotron radiation at the HE-SGM beamline. We acknowledge computational resources from HRZ Marburg.

## Author contributions

M.D. and G.W. conceived this project. M.D. performed all STM and Kelvin probe measurements. M.D. and P.M.D. performed the NEXAFS measurements. P.M.D. developed the code for simulations of intermolecular interactions and molecular dynamics. M.D. used this code to simulate molecular attachment processes. P.M.D. performed all other computational work and analysed the boundary free energy of nanosheets. M.W.T. and N.M. synthesised L- and M-F6PEN. U.K. supervised the synthesis. The paper and the Supporting Information were written by P.M.D., M.D. and G.W.

## Funding

## Competing interests

The authors declare no competing interests.
