## [Peer Review File · Nature Communications]

Shape control in 2D molecular nanosheets by tuning anisotropic intermolecular interactions and assembly kineticsReviewers' Comments:

Reviewer #1:

Remarks to the Author:

In the manuscript, (sub)monolayers of fluorinated pentacene derivatives were deposited on MoS₂ crystals using the experimental setup previously published by the group (e.g. ref 30) and imaged using scanning tunnelling microscopy (STM). STM micrographs show different structural motifs of the molecular layers when formed by adsorption or partial desorption. The interpretation of the observed differences relies on classical molecular dynamics (MD) simulations. The paper is well written and has attractive figures. However, the computational evidence is not sufficient to support the observations and conclusions.

In MD modelling, as described, interactions with MoS₂ substrate are not included, and it is argued that molecule – substrate interactions are exceptionally weak so that a single molecular layer can be treated as free standing. There is no obvious evidence for such a statement, both experimentally and computationally. If the substrate lattice constant does not matter much for the formation of an ordered adsorbate layer, one may expect similarly structured monolayers to be formed on other atomically flat and rather inert solid surfaces. It is however well known that details of the packing, including the symmetry of the adsorbate unit cell, depend on the substrate since they are determined by very small differences in the free energy. The effect of the substrate should be discussed, both experimentally and theoretically, and “self-assembly on surface” should feature in the title.

Most of the computational reasoning, starting with Figure 2, is based on a “brick-by-brick” mechanism describing the energetics (and probability) of favourable attachment/desorption sites. Self-assembly is rarely driven by the initial attachment of a single molecule – it is inherently a process driven by the collective behaviour. This collective self-assembly behaviour needs to be captured in computational modelling to strengthen the conclusions of the paper.

As presented, the simulations of growth kinetics are not convincing. In figure 2b, small cusps of weak attraction along the SE are not accessible due to strong repulsion barrier at longer separation distances and hence cannot simply promote the growth of the layer at the SE following this “brick-by-brick” mechanism.

I really liked the results examining ad(de)sorption of a half-fluorinated pentacene derivative with distinctly bipolar electrostatic behaviour but the lengthy repetition of computational results for this interesting case is distracting.

Other more technical queries:

Why are the computational results obtained with an unknown, “self-programmed” code? Why can they not be repeated with existing MD codes available to academic use?

Parameters for F-F interactions are too old.

Taking the geometric mean value for the interaction parameters of mixed cases (C-F, N-C etc) is a very crude approximation.

How does the value for the reaction rate constant of the detachment (from nanosheets) depend on the pre-factor value for desorption, which is defined by the surface dynamics?

Reviewer #2:

Remarks to the Author:

In this work, the Authors perform an experimental/computational investigation of a system of organic molecules - fluorinated pentacene derivatives (L-F6PEN and M-F6PEN) - on a MoS₂ (001) substrate. Different hydrogen substitutions with fluorine produce different asymmetric electrostatic potentials around the molecules, which is key to the observed assemblies. The molecule-substrate (M-S) interaction is assumed to be weak, and then neglected in the modelling, and the surface metallicity is enough to perform a STM investigation. After an annealing at T \approx 400 K, the two molecules assemble in islands of different shape: L-F6PEN islands tend to be vertically elongated (along a direction denoted LE); M-F6PEN islands are essentially isotropic. Free energy considerations lead the Authors to conclude that thermodynamically stable islands would be elongated along a different direction (SE), and thus that the shape of the islands experimentally observed is kinetically controlled.

Please see a list of points below (not in order of importance):

- Fig. 1a: for the L-F6PEN molecule, the STM signal seems stronger on the hydrogen side (instead of the fluorine side) as the molecular shape in the inset is slightly asymmetric. In the M-F6PEN case (Fig. 4a) it is the opposite, a much stronger STM signal is in correspondence with the fluorine area, giving a strong zig-zag signal. The Authors should elaborate on these facts and possibly give an explanation.

-Fig. 1b: In the text/caption, it would be better to refer explicitly to the Computational section to understand how the electrostatic potential is obtained.

-Page 1, col 2: "...we find an oblique unit cell with...". If this cell corresponds to the blue unit cell in Fig. 1c, this needs to be indicated explicitly.

-Page 2, col 1: "Submonolayer films with a nominal thickness of 0.8Å (cf. Fig. 1d)". It is not specified how this thickness is obtained and what is the relation with Fig. 1d (not showing any information about thickness).

-Page 2, col 1: "...at 400K for 1min to activate partial desorption of molecules...". The Authors should specify if desorbed molecules were detected in the chamber, or desorption is only assumed by a reduction of molecular average density on the substrate. Further in the text, they seem to exclude completely that annealing could make the molecules access the state in Fig. 1e (that must represent an energy local minimum, deeper than the one in Fig. 1d) or other configurations, even without desorption. An elaboration on these points should be made (see also below).

-Page 2, col 1: "SE (SE-extended cf. Fig. 1c)". This SE/LE and SE/LE-extended terminology is a bit misleading. The Authors should state explicitly (if it is the case) that LE/SE represent edges/sites, LE/SE-extended represent directions.

-Fig. 1c: it should be clarified if the 4x2 super cell has a specific meaning or is just an example.

-Page 2, col 1: "The white area... due to mutual overlap of interacting molecules." By "overlap" it is meant a steric effect and/or Pauli repulsion? Moreover, from Figs. 2 a and b the excluded area seems huge and the molecules well inside the island seem not surrounded by this excluded area. The meaning of excluded area need to be clarified.

-Fig. 2 caption: all symbols/abbreviations (e.g. n_{se}, etc.) should be described also in the caption. Moreover, the very right panel needs to be introduced.

-Fig. 2: The discussion about the mechanisms, in terms of interactions, leading to the p(LE) anisotropy should be extended. The Authors could consider including part of Supporting Information Section 5 in the main text.

-Fig 3: A clear distinction between "desorption" and "detachment" should be made. Detachment could

happen from an island, without desorption.

-Page 3, col 1: "Molecules at the SE lack only the weaker nearest-neighbour bond whereas LE molecules lack a strongly attractive neighbour." This sentence is not clear. What are "LE molecules?" If LE is an edge, all molecules have it. Moreover, how can the cohesive energy coming from a stronger interaction (of two molecules facing each other along the LE) be smaller than the cohesive energy fraction coming from interaction of two molecules facing each other along the SE? Unless other effects are included, i.e. the position of the molecule with respect to the corners/edge the island.

-Page 4, col 1: "...interactions along the top SE are almost exclusively attractive and therefore have a longer range despite overall weaker attraction." Is a top SE different from a bottom one? Are other interactions along other directions (e.g. LE) repulsive instead? A clarification of these points is needed.

-Page 5, col 1: "using MD simulations based on an atomistic model and fundamental vdW force fields." Here and in some other sentences (e.g. in the abstract) it is written that only vdW M-M interactions are considered. In other parts (the Computational section/Supporting Information) electrostatic interactions and short range Pauli repulsion are included. This needs to be corrected, using coherent statements.

-Supporting information, Section 2: considering only M-M interactions seems a crude approximation, as M-S forces may constitute a key element in molecular diffusion (energy barriers, etc.) especially considering kinetics-controlled processes. The fact that no M-S charge transfer is found, as implied by an unchanged work-function upon molecular deposition does not exclude, in principle, a surface corrugation influencing the assembly and paths to the thermodynamic energy minimum. I think the Authors should provide further arguments for their choice to not include a M-S interaction in their calculations.

-Page 5, col1 and Supporting Information Sec 7, connected to previous point: a model is presented to find the optimal shape of the islands at fixed number of molecules (NoM). The optimal shape found contrasts the experimental results. With fixed NoM, the desorption is not considered. I wonder if this should, instead, be a parameter in the model, allowing the NoM to change. Moreover, have the Authors thought to perform annealing experiments at different temperatures, lower and higher than ~400K? In the work the M-S interaction is completely neglected, but a surface adsorption energy is present, thus it would be worth to check if annealing at different T could produce a molecular rearrangement (towards a deeper energy minimum), without desorption. A SE-extended shape could be found, better validating the claim of kinetic control.

-The "References" section is divided in two parts, with the Methods section in between, this formatting should be corrected.

-In the abstract/conclusions, the Authors describe theirs as a novel "approach." Although the work is interesting, it is not exactly clear that is meant by "approach." Moreover, recent studies - e. g. Commun Chem 1, 66 (2018) - analysed already in detail the possibility to use kinetics to increase the number of metastable self-assembled molecular network on substrates.

In summary, the manuscript is interesting, the quality of the analysis is overall valid, even if some strong approximations are adopted. The level of the presentation could be improved in some parts, e.g. the SE/LE terminology could be clearer. More explicit descriptions (e.g. in captions) would be beneficial. Sometimes there is a not an in-depth discussion on the results in the main text, especially concerning the electrostatic interactions at play, but this is complemented by the Supporting Information. Still, if the format allows, I would include in the main text some considerations present in the SI.

In my opinion the results found, even if interesting, are specific to the field. The idea of using kinetics

to obtain self-assemblies is not new and I am not sure if the sets of rules defined at the end could be effectively used for other systems, to control the shape of self-assembled islands (this is, in general, an extremely challenging task). The impact of the presented results makes the manuscript more suitable to other journals. For all these reasons, I do not recommend publication in Nature Communications. Alternative choices, after addressing all points suggested above, could be Communications Chemistry/Physics or Scientific Reports.

Reviewer #3:

Remarks to the Author:

This work proposed a design strategy of controlling 2D organic monolayer on TDMC taking advantages of anisotropic Van de Waals interactions between functionalized acenes. STM was used to monitor the adsorption and desorption of nanosheets, and different shapes were observed accordingly. MD and MC simulations were performed for the attachment to and detachment from the nanosheet, respectively. And it was concluded that the desorption process is kinetic controlled because the free energy unfavorable LE extension shape is dominated. This is an interesting work, however the method used is not clarified and the analysis is confusing. I cannot recommend it for publication in its present form. Comments are given below.

1. Since the author claimed that MD results supported their conclusions, the details have to be provided for MD simulations, including the setup of simulation box, the algorithm, the ensemble used, etc. otherwise it is impossible to judge if the results obtained are reliable or not. Note that the MC simulations used for the detachment process is different from MD simulations.

2. When modeling the TDMC/OSC complex in simulations, the interaction between monolayer and the substrate was not included. Although for a single organic molecule, the intermolecular interactions could be much larger than the molecule/substrate interaction, this may not be true for larger aggregate. Also functionalized acene may have stronger interactions with the substrate than unfunctionalized one, and surface defects and/or reconstruction may also result in nonnegligible molecular substrate interactions. Systematic evaluation of the effect of molecule substrate interaction on the shape control is recommended.

3. It is argued that the sheet growth was governed by the nearest neighbor interaction, however, only the single molecule process was discussed. It is possible that the entire sheet breaks into several parts, and when the entropy contribution was included (the smaller sheet can rotate especially at elevated temperatures), the desorption process may actually be thermodynamic controlled. Indeed Figure 1e displays several sheets appearing in different orientations. Large scale STM images for the other functionalized molecule are helpful for further analysis.

I would also recommend taking the STM images at different times to monitor the adsorption/desorption processes (if possible).

Reviewer #4:

Remarks to the Author:

The authors show that two-dimensional (2D) nanosheets of regioselectively fluorinated pentacene derivatives exhibit different shapes when formed by adsorption or desorption. They use molecular dynamics (MD) simulations based on an atomistic model and fundamental van der Waals (vdW) force fields to explain the kinetically controlled formation of 2D molecular nanosheets. They expand the approach to mesoscopic structural control of organic molecular nanosheets on molybdenum disulphide. I recommend to publish after the following questions to be addressed.

1. The STM micrographs of nominal monolayer and islands of L-F6PEN had better exhibit in same range. Fig. 1d show much worse quality and should be updated. The scale bars of inner figures are missed. These islands of L-F6PEN in Fig. 1d show different orientation, but they were in same

orientation before annealed from the complete monolayer, the simulation of desorption kinetics in figure 3 not mention the phenomenon as well, could the authors give some explains?

2. As showed in Fig. 1d, submonolayer films with a nominal thickness of 0.8\AA form well-ordered islands. The authors had better offer some statistical data of these islands, containing the area and aspect ratio. The STM micrographs and statistical data of different nominal thickness should also be showed.

3. Fig. 3b shows a map of the average desorption sequence of molecules from an L-F6PEN nanosheet of $mLE=20$ by $nSE=50$ molecules. Why did the authors choose $20*50$ nanosheet? What is typical shape and aspect ratio of complete monolayer?

4. The authors claim that their novel approach to structural control in 2D molecular nanosystems can be applied to a vast number of molecular materials on weakly interacting substrates. Could they offer some examples in simulation or experiment?

REVIEWER COMMENTS

Reviewer #1 (Remarks to the Author):

In the manuscript, (sub)monolayers of fluorinated pentacene derivatives were deposited on MoS₂ crystals using the experimental setup previously published by the group (e.g. ref 30) and imaged using scanning tunnelling microscopy (STM). STM micrographs show different structural motifs of the molecular layers when formed by adsorption or partial desorption. The interpretation of the observed differences relies on classical molecular dynamics (MD) simulations. The paper is well written and has attractive figures. However, the computational evidence is not sufficient to support the observations and conclusions.

In MD modelling, as described, interactions with MoS₂ substrate are not included, and it is argued that molecule – substrate interactions are exceptionally weak so that a single molecular layer can be treated as free standing. There is no obvious evidence for such a statement, both experimentally and computationally.

We thank the reviewer for this remark, as it shows that some of our arguments regarding the weak molecule-substrate interactions should be emphasised and discussed in more detail. Direct experimental evidence for the weakness of molecule-substrate interaction potential for pentacene derivatives adsorbed on MoS₂(001) substrates, both in absolute magnitude and spatial corrugation, is provided in Refs. [Kachel et al., Chem. Sci., 12, 2575 (2021)], [Tumino et al., J. Phys. Chem. C, 126, 1132 (2022)], and [Dreher et al., Chem. Mater., 32, 20, 9034 (2020)].

In Ref. [Kachel et al., Chem. Sci., 12, 2575 (2021)], the molecule-substrate binding energy between MoS₂ and two close relatives of the F₆PENs, pentacene (PEN) and perfluoropentacene (PFP), is measured directly using temperature-programmed desorption (TPD). It is shown that this energy is almost 50% smaller than corresponding values for Au(111) substrates [cf. Dombrowski et al., Nanoscale, 13, 13816 (2021)]. Moreover, an analysis of the entropy of the system through TPD in combination with STM experiments shows that the molecules adsorbed on MoS₂ are exceptionally mobile even at cryogenic temperatures, leading to the conclusion that these systems, in contrast to the same molecules adsorbed on metallic substrates [e.g. Dombrowski et al., Nanoscale, 13, 13816 (2021)], do not form ordered layers due to a weak corrugation of the molecule-substrate interaction potential.

Tumino *et al.* show that, upon deposition onto flakes of 2D MoS₂ on a supporting Au(111) substrate, PEN aggregates at the edges of the MoS₂ flakes and thus on the SiO₂ substrate, rather than on top of the MoS₂ flakes. DFT calculations show that this is caused by a weak molecule-substrate interaction on the MoS₂ flake, causing diffusion towards more stable sites on the Au(111) surface. The DFT results are in good agreement with experimental data from Kachel *et al.* and Dombrowski *et al.*

Ref. [Dreher et al., Chem. Mater., 32, 20, 9034 (2020)] shows the consequences of such a weak molecule-substrate interaction for structure formation, which is essentially governed by intermolecular interactions. The observed point-on-line epitaxy of PFP films grown on MoS₂ indicates that, upon film growth, the orientation of the film on the substrate is optimised under the condition of an optimum intermolecular arrangement. By contrast, a significant corrugation of the molecule-substrate interaction potential would result in a commensurate or higher order commensurate molecular superstructure, as, for instance, observed for PEN and PFP on Au(111) and Ag(111) substrates. [Wang et al., Nanoscale Adv., 3, 9, 2598 (2021)] [Götzen et al., Langmuir, 27, 3, 993 (2011)] [D.B. Dougherty et al. J. Phys. Chem. C 2008, 112, 51, 20334–20339].

While this is no direct proof for the partially fluorinated PENs used in this work, other studies with partially fluorinated acenes [Hofmann et al., Angew. Chem. Int. Ed., 59, 16501 (2020)] have shown that optoelectronic properties lie between those of PEN and PFP, so that one can make at least a strong argument that this also applies to molecule-substrate interaction energies.

We have emphasised and explained in more detail the above arguments in our manuscript and the methods section, hence justifying that the presented adlayers on MoS₂ can be well described as quasi freestanding layers, while such a description would not be possible on metal substrates.

If the substrate lattice constant does not matter much for the formation of an ordered adsorbate layer, one may expect similarly structured monolayers to be formed on other atomically flat and rather inert solid surfaces. It is however well known that details of the packing, including the symmetry of the adsorbate unit cell, depend on the substrate since they are determined by very small differences in the free energy. The effect of the substrate should be discussed, both experimentally and theoretically, and “self-assembly on surface” should feature in the title.

In continuation of our reply above, we strongly disagree with the reviewer’s critique. We do not argue that molecule-substrate interactions can generally be neglected. In particular in the well-established field of metal-organic interfaces, it is known the spatial corrugation of the molecule-substrate interaction potential often determines structure formation of organic adlayers. However, as discussed in the previous comment, in the special case of MoS₂ and other TMDCs, this interaction, both in its absolute strength and spatial corrugation, is exceptionally weak. In addition to a weak molecule-substrate interaction, intermolecular interactions are exceptionally strong for our molecules. This further shifts the balance of molecule-substrate and intermolecular

interactions to a point where the former is negligible. For a single molecule on the MoS₂ surface, the corrugation of the molecule-substrate interaction potential is likely to lead to a preferential azimuthal orientation of the molecule. Nevertheless, the substrate does not determine molecular geometries, at least for similar acenes on MoS₂.

To distinguish our findings from the field of self-assembled monolayers (SAMs), we changed our title from 'Shape Control of 2D Molecular Nanosheets through Kinetics of Self-Assembly' to 'Shape Control of 2D Molecular Nanosheets through Kinetics of their Assembly'

Most of the computational reasoning, starting with Figure 2, is based on a "brick-by-brick" mechanism describing the energetics (and probability) of favourable attachment/desorption sites. Self-assembly is rarely driven by the initial attachment of a single molecule – it is inherently a process driven by the collective behaviour. This collective self-assembly behaviour needs to be captured in computational modelling to strengthen the conclusions of the paper.

We agree with the reviewer that simulations of the growth of nanosheets and thus the true self-assembly process cannot be based on a simple brick-by-brick mechanism. However, this is not our goal, as we already know the results of the assembly process from our experimental data. We are rather interested in understanding growth mechanism through interaction potentials and the statistical analysis of attachment and desorption processes. Naturally, the simulation of the attachment of a single molecule to a rigid nanosheet does not give detailed insight into the kinetic growth of a nanosheet from ground up. However, through the statistical analysis, it does tell us where molecules are more likely to attach, which might provide more insights into the growth mechanisms than an exact molecular dynamics simulation of a nanosheet.

For the desorption simulation, we have arguments for the validity of a "brick-by-brick" mechanism that are discussed in detail in our response to the reviewer's final comment. In short, the strong lateral intermolecular interactions, coupled with an exceptionally weak molecule-substrate bond, render the detachment of molecules from the sheet without immediate desorption highly unlikely. The experimental observation of different preferential nanosheet extensions upon growth and annealing, the latter leading to island shapes that are energetically unfavourable, supports the argument that no detachment of molecules from a nanosheet without immediate desorption due to the elevated substrate temperature is possible. Hence, the "brick-by-brick" model is also justified for the desorption simulations.

As presented, the simulations of growth kinetics are not convincing. In figure 2b, small cusps of weak attraction along the SE are not accessible due to strong repulsion barrier at longer separation distances and hence cannot simply promote the growth of the layer at the SE following this "brick-by-brick" mechanism.

We are grateful for this remark, which shows that our discussion has not been as clear as it should. Of course, repulsive barriers must be overcome to reach the above-mentioned cusps of attractive interaction that allow for attachment of molecules to a nanosheet. However, such barriers can be overcome with sufficient kinetic energy; after all, growth of molecular structure is a kinetic process. We would like to emphasise that other well-studied molecules such as PEN or copper phthalocyanine interact purely repulsively at distances exceeding few Å, and only at very close distances interact attractively due to dispersion forces [Kröger et al., *New J. Phys.*, 12, 083038 (2010)] and [Dombrowski et al., *Nanoscale*, 13, 13816 (2021)]. Following the reviewer's argument, such molecules could not follow a 'brick-by-brick' growing mechanism, unless a sufficiently strongly interacting substrate is found that forces molecules to come close as the surface coverage approaches a complete monolayer. However, scanning tunnelling microscopy data of submonolayers show clusters of these molecules, which could not be formed if repulsion could not be overcome by kinetics to reach the attractive binding regime. Moreover, for the systems used in this work, there is significant electrostatic attraction. Hence, there is by no means a 'strong repulsion barrier', but a slight repulsive barrier that can be overcome kinetically. We have modified our discussion to explain this concept in more detail.

I really liked the results examining ad(de)sorption of a half-fluorinated pentacene derivative with distinctly bipolar electrostatic behaviour but the lengthy repetition of computational results for this interesting case is distracting.

We thank the reviewer for this remark and have shortened the discussion of the computational results to make it more concise.

Other more technical queries:

Why are the computational results obtained with an unknown, "self-programmed" code? Why can they not be repeated with existing MD codes available to academic use?

The latter question is somewhat surprising, since it is stated nowhere in our submission that our results cannot be replicated with other MD code. Other MD codes were simply not tested, since no different results are to be expected. We have clearly stated all of our fundamental assumptions and strongly believe that any MD code will replicate our results if those assumptions are implemented.

We chose to write our own code instead of using existing codes for maximum flexibility and understanding of the inner workings of the code. Had we used existing MD software, we would have had to write our own code for the attachment and TPD simulations anyways. We regret to not have submitted our code together with the initial submission of the manuscript and hope that our submission now will resolve the reviewer's doubts. We have uploaded our fully documented source code to an open access repository.

Parameters for F-F interactions are too old.

We are grateful for this remark. However, these parameters concern only the dispersion forces and Pauli repulsion, which (with the exception of Pauli repulsion at extremely close distances) are much weaker than the electrostatic interactions that do not rely on this parameter but are computed from the NBO charges. Therefore, even if more modern parameters could increase our computational accuracy, the gain in accuracy is unlikely to be significant in view of the small contribution of the dispersion forces to long range interaction in comparison to Coulomb forces. For the kinetic simulations, we do not expect notable changes by updating this material parameter due to the rather insignificant contribution of the dispersion interactions. Only intermolecular distances in the optimised molecular structures might vary slightly due to changes in the Pauli repulsion, but the packing motif would not be affected in any significant way since it is predominantly determined by the anisotropic electrostatic interactions. Furthermore, we would like to argue that the age of parameters is not necessarily correlated to their accuracy. For all of the above reasons, we have decided to not update our simulations with a more modern parameter for F-F dispersion and Pauli interactions.

Taking the geometric mean value for the interaction parameters of mixed cases (C-F, N-C etc) is a very crude approximation.

These interaction parameters concern only dispersion and Pauli interaction. As discussed above, corrections in the dispersion forces barely affect our results. Moreover, this method has already been applied elsewhere [e.g. Kröger *et al.*, J. Chem. Phys., 135, 234703 (2011)] to compute structural configurations of molecules that are in good agreement with experimental data.

How does the value for the reaction rate constant of the detachment (from nanosheets) depend on the prefactor value for desorption, which is defined by the surface dynamics?

We thank the reviewer for this remark, which shows that the distinction between detachment has not been sufficiently clear in our initial submission. To model detachment as it occurs in our experiments, we have simulated temperature-programmed desorption experiments, hence the discussion of the prefactor of desorption. We have used the terms 'detachment' and 'desorption' synonymously, which is inaccurate and therefore has caused confusion. We do not consider detachment from a nanosheet without desorption or any form of surface dynamics during the desorption experiments. However, due to strong intermolecular binding and its important contribution to the stability of the molecular film at elevated temperatures, it is unlikely that molecules can detach from a nanosheet without immediate desorption, as intermolecular interactions are of the same order of magnitude as or even larger than the molecule-substrate bond (cf. Fig. S6 in the revised *Supp. Inf.*). Therefore, detachment from a sheet therefore leads to an immediate decrease of the activation energy of desorption by more than 50%. Therefore, we believe that the reaction constant for detachment is directly related to the prefactor of desorption, to which the reaction rate constant for desorption is directly proportional. After all, detachment from a nanosheet is directly related to surface dynamics.

Our experiments further show no variation of the nanosheet shape if different annealing temperatures are used. If structural reordering, i.e., detachment of molecules from a sheet and subsequent attachment elsewhere would be possible, this should lead to a change of nanosheet shapes to an energetically more favourable configuration that is experimentally not observed, thus supporting our energetic argument.

We have modified the manuscript accordingly to clarify this assumption of our simulations and the distinction between detachment and desorption. Furthermore, we have discussed the surface dynamics in more detail.

Reviewer #2 (Remarks to the Author):

In this work, the Authors perform an experimental/computational investigation of a system of organic molecules - fluorinated pentacene derivatives (L-F6PEN and M-F6PEN) - on a MoS₂ (001) substrate. Different hydrogen substitutions with fluorine produce different asymmetric electrostatic potentials around the molecules, which is key to the observed assemblies. The molecule-substrate (M-S) interaction is assumed to be weak, and then neglected in the modelling, and the surface metallicity is enough to perform a STM investigation. After an annealing at T~400 K, the two molecules assemble in islands of different shape: L-F6PEN islands tend to be vertically elongated (along a direction denoted LE); M-F6PEN islands are essentially isotropic. Free energy considerations lead the Authors to conclude that thermodynamically stable islands would be elongated along a different direction (SE), and thus that the shape of the islands experimentally observed is kinetically controlled.

Please see a list of points below (not in order of importance):

- Fig. 1a: for the L-F6PEN molecule, the STM signal seems stronger on the hydrogen side (instead of the fluorine side) as the molecular shape in the inset is slightly asymmetric. In the M-F6PEN case (Fig. 4a) it is the opposite, a much stronger STM signal is in correspondence with the fluorine area, giving a strong zig-zag signal. The Authors should elaborate on these facts and possibly give an explanation.

In this point we disagree with the reviewer. The inset in Figure 1a clearly shows an asymmetric shape of the molecule, as the reviewer stated. The brightest area is slightly shifted to the right with respect to the geometrical centre of the molecule and hence towards the fluorinated side, as we point out in the scheme.

-Fig. 1b: In the text/caption, it would be better to refer explicitly to the Computational section to understand how the electrostatic potential is obtained.

We thank the reviewer for this remark and have added references to the computational section in the manuscript.

-Page 1, col 2: "...we find an oblique unit cell with...". If this cell corresponds to the blue unit cell in Fig. 1c, this needs to be indicated explicitly.

We thank the reviewer for this remark. The assumption that the unit cell in question is the one shown in blue in Fig. 1c is correct. We have added a reference to the figure in question to make this clearer.

-Page 2, col 1: "Submonolayer films with a nominal thickness of 0.8Å (cf. Fig. 1d)". It is not specified how this thickness is obtained and what is the relation with Fig. 1d (not showing any information about thickness).

We are grateful for the reviewer's remark, highlighting an inconsistency in our description of film thicknesses / surface coverages. We have extended our discussion of thickness determination (and thus coverage determination) during film growth in the experimental section and added coverages for grown films in figure captions and the *Supp. Inf.*

From X-ray standing wave (XSW) measurements (Wang et. al, *Nanoscale Adv.* 3, 2598-2606 (2021)) the height of a monolayer of flat-lying (perfluoro-)pentacene molecules is determined to be around 3.3Å. We checked with STM for L-F₆PEN and M-F₆PEN films with a nominal thickness of that value, that a complete monolayer is formed. Since STM only images a tiny area of the whole film, we carefully examined the sample at different positions to check, that there is no distinct thickness gradient over the sample. Using this as calibration, we determined the nominal thickness of the submonolayer films by a QCM. Now we give the nominal film thickness for a monolayer which, in our experience, amounts to 3.0Å.

-Page 2, col 1: "...at 400K for 1min to activate partial desorption of molecules...". The Authors should specify if desorbed molecules were detected in the chamber, or desorption is only assumed by a reduction of molecular average density on the substrate. Further in the text, they seem to exclude completely that annealing could make the molecules access the state in Fig. 1e (that must represent an energy local minimum, deeper than the one in Fig. 1d) or other configurations, even without desorption. An elaboration on these points should be made (see also below).

We thank the reviewer for this remark. Desorbing molecules cannot be detected with the system that was used for our experiments. However, we do not simply assume that desorption has taken place. For the submonolayer films that were fabricated by means of partial desorption, initial films were created with coverages exceeding a complete monolayer. Before partial desorption of a monolayer, completeness of the layer was checked thoroughly by means of STM at different positions on the sample surface to assure that any observation of a submonolayer can only be caused by partial desorption. The initially deposited surface coverage was checked through rate determination by means of a quartz crystal microbalance.

Regarding the second point of this comment, we would first like to point out that the state shown in Fig. 1d is closer to the energy local minimum (with a short LE edge) than the state in Fig. 1e. This serves as direct proof

that detachment without desorption is not possible, since the state in Fig. 1e was achieved by annealing and thus adding energy to the system, which should promote detachment and diffusion if it was possible. The finding of a state that deviates strongly from the energy local minimum after annealing shows that reorganisation is not possible. We have modified the manuscript to explain this argument further and emphasise it more strongly.

-Page 2, col 1: "SE (SE-extended cf. Fig. 1c)". This SE/LE and SE/LE-extended terminology is a bit misleading. The Authors should state explicitly (if it is the case) that LE/SE represent edges/sites, LE/SE-extended represent directions.

We are grateful for this remark and updated the terminology in the text. Since the extension of a sheet is in the ideal case in a direction of the unit cell vectors of the adlayer, we now denote the extension as $\langle \vec{a} \rangle$ - and $\langle \vec{b} \rangle$ -extended sheets according to the directions of the unit vectors of the molecular superstructure. With this terminology it should be clear in which crystallographic direction the sheets are extended.

Attachment and desorption occur not in a direction, but at a location. Therefore, we decided to denominate this process as, for instance, 'attachment at the SEs' or 'desorption from the LEs'. Thus, the actual edges should not be mixed up with the direction of extension of the sheet. We agree with the reviewer, that it can be confusing for the reader to have one hand the shape of an island, and on the other hand the inner structure of it in mind. These two aspects should be disjunct by this wording now.

-Fig. 1c: it should be clarified if the 4x2 super cell has a specific meaning or is just an example.

The depicted 4x2 super cell has no specific meaning and was chosen arbitrarily. We have modified the figure caption to clarify this.

-Page 2, col 1: "The white area... due to mutual overlap of interacting molecules." By "overlap" it is meant a steric effect and/or Pauli repulsion? Moreover, from Figs. 2 a and b the excluded area seems huge and the molecules well inside the island seem not surrounded by this excluded area. The meaning of excluded area need to be clarified.

We are thankful for this valuable remark. The origin of the excluded area lies in Pauli repulsion. Due to its exponential increase at short intermolecular distances, Pauli repulsion defines a rather sharp boundary that cannot be penetrated by other molecules. Since Fig. 2b shows the interaction potential of a single molecule with the island depicted in the centre of the excluded area, the depicted area is a property of the island caused by the interaction of molecules at the island edge with a probe molecule. The size of these areas is directly related to the size of the molecules: As depicted in the inset between Figs. 2a and b, the excluded area marks all centre-to-centre distances of the probing molecule and its surrounding (single molecule in 2a or island in 2b). The reviewer's confusion regarding the size of the excluded areas in Fig. 2 might be caused by Fig. 1b, where a similar white area is shown around the molecule that is much smaller. Here, the white area is technically not an excluded area, since the probing particle is an electron (to only probe electrostatic interactions) rather than a molecule. Since there is no Pauli repulsion included in this calculation, we have simply cut off the energy scale to approximately achieve a white area of the size of the van der Waals box of the depicted molecule.

To avoid such confusion, we have explained the exclusion area in more detail and discuss the differences between the white area in Figs. 1b and 4a and what we define as excluded areas in Figs. 2a, 2b and 4d.

-Fig. 2 caption: all symbols/abbreviations (e.g. n_{se} , etc.) should be described also in the caption. Moreover, the very right panel needs to be introduced.

We are grateful for this remark and have modified the caption accordingly.

-Fig. 2: The discussion about the mechanisms, in terms of interactions, leading to the p(LE) anisotropy should be extended. The Authors could consider including part of Supporting Information Section 5 in the main text.

We thank the reviewer for this remark and have extended the discussion in the main paper. However, since another reviewer criticised the lengthy discussion of computational results, we have decided against including major parts of the discussion from the *Supp. Inf.* since we already extended our discussion by adding another molecule that allowed us to discuss the kinetics of structure formation and the mechanisms behind them in more detail.

-Fig 3: A clear distinction between "desorption" and "detachment" should be made. Detachment could happen from an island, without desorption.

We are grateful for this remark. Unfortunately, the terms have been used synonymously, which is inaccurate. However, based on the computed interaction energies and STM experiments with monolayers that were annealed at different temperatures, we argue that detachment of a molecule from a nanosheet without its immediate desorption is highly unlikely if not impossible. Upon partial desorption of monolayers of L-F₆PEN, we always find $\langle \vec{a} \rangle$ -extended nanosheets independent of the annealing temperature (provided that it is high enough to activate desorption of molecules and low enough to avoid complete desorption of the monolayer).

Only very small nanosheets formed upon significant desorption (cf. Fig. 1e) deviate from this anisotropic extension, which is consistent with our simulations that do not allow detachment without immediate desorption. Therefore, we can conclude that molecules are highly unlikely to detach from a nanosheet and attach elsewhere, which would result in changes of the nanosheet shape.

This is caused by the strong lateral intermolecular binding (cf. cohesive energy maps in Fig. S6) together with a weak molecule substrate bond. For an L-F₆PEN molecule in the centre of a nanosheet, the cohesive energy is approximately 170 kJ/mol, which is significantly larger than the molecule substrate bond of about 130 kJ/mol [Kachel et al., Chem. Sci., 12, 2575 (2021)]. Therefore, once a molecule detaches from a nanosheet, its activation energy of desorption decreases by more than ~50%, which is likely to result in immediate desorption at the temperature required to detach molecules.

We have modified the relevant discussion to make a clearer distinction between desorption and detachment and explain our argument that they are essentially the same in our case.

-Page 3, col 1: "Molecules at the SE lack only the weaker nearest-neighbour bond whereas LE molecules lack a strongly attractive neighbour." This sentence is not clear. What are "LE molecules?" If LE is an edge, all molecules have it.

We thank the reviewer for this remark, highlighting unclear phrasing in our original manuscript. The term "LE molecules" is supposed to refer to molecules located at the edge labelled LE, i.e., the edge at which molecules have no neighbours at their LE. As discussed above, we have modified our terminology to clarify such statements.

Moreover, how can the cohesive energy coming from a stronger interaction (of two molecules facing each other along the LE) be smaller than the cohesive energy fraction coming from interaction of two molecules facing each other along the SE? Unless other effects are included, i.e. the position of the molecule with respect to the corners/edge the island.

Again, it appears that our terminology has not been sufficiently clear. It is correct that interactions between molecules facing each other along the LE are stronger than for an SE configuration. Therefore, the cohesive energy of the former is not smaller than the latter. What we want to describe here is that molecules at the LE-exposed edge of the island are missing one LE-facing neighbour (considering only nearest neighbours at the four molecular edges), whereas molecules at the SE-exposed edge of the island are missing an SE-facing neighbour. Therefore, the cohesive energy is smaller at the LE-exposed edge, and desorption is most likely to occur here.

Due to the obvious confusion of our discussion here, we have modified it to improve clarity.

-Page 4, col 1: "...interactions along the top SE are almost exclusively attractive and therefore have a longer range despite overall weaker attraction." Is a top SE different from a bottom one?

In this particular case, there is a difference due to the alternating orientation of M-F₆PEN molecules and the different relative strength of mutual repulsion of oppositely oriented molecules, depending on whether they are facing with their H-terminated SEs or their F-terminated SEs. Therefore, there is a slight difference in the attachment energy maps of M-F₆PEN in the configuration shown in Fig. 4d (independent of island dimensions). We have reworded parts of the discussion to avoid confusion.

Are other interactions along other directions (e.g. LE) repulsive instead? A clarification of these points is needed.

Unfortunately, we do not understand this question. As one can see in Fig. 4d, interactions along the LE-exposed edge of the island alternate between attractive and repulsive regions. Fig. S4 shows further attachment energy maps for different orientations of the probing admolecule.

It appears that this discussion is not perfectly clear. Therefore, we have modified this discussion to improve clarity.

-Page 5, col 1: "using MD simulations based on an atomistic model and fundamental vdW force fields." Here and in some other sentences (e.g. in the abstract) it is written that only vdW M-M interactions are considered. In other parts (the Computational section/Supporting Information) electrostatic interactions and short range Pauli repulsion are included. This needs to be corrected, using coherent statements.

We thank the reviewer for this remark. The term 'vdW interactions' refers to a range of interactions that includes electrostatic interactions, Pauli repulsion and dispersion forces. We agree that the cited sentence is not sufficiently clear and have modified it as well as other similar sentences in the main paper and the *Supp. Inf.*

-Supporting information, Section 2: considering only M-M interactions seems a crude approximation, as M-S forces may constitute a key element in molecular diffusion (energy barriers, etc.) especially considering kinetics-controlled processes. The fact that no M-S charge transfer is found, as implied by an unchanged work-function upon molecular deposition does not exclude, in principle, a surface corrugation influencing the

assembly and paths to the thermodynamic energy minimum. I think the Authors should provide further arguments for their choice to not include a M-S interaction in their calculations.

We agree that, generally, this is a crude approximation. However, other studies show that:

- i) Pentacene (PEN) and perfluoropentacene (PFP) are highly mobile on MoS₂, indicating that the corrugation of the M-S interaction potential is relatively weak (in addition to an exceptionally weak binding energy of around 130 kJ/mol, compared to, for instance, 220 kJ/mol for pentacene on Au(111)) [Kachel et al., Chem. Sci., 12, 2575 (2021)].
- ii) Pentacene deposited on 2D MoS₂ flakes on Au(111) substrates does not stay on the MoS₂ flakes, as it prefers to adsorb on the Au(111) substrate. [Tumino et al., J. Phys. Chem. C, 126, 1132 (2022)] DFT calculations show that the M-S interaction with MoS₂ is significantly weaker than that with Au(111), in agreement with the experimental reports by Kachel *et al.* and Ref. [Dombrowski et al., Nanoscale, 13, 13816 (2021)].
- iii) Films of PFP on MoS₂ exhibit a point-on-line epitaxy, showing that structure formation is first and foremost determined by intermolecular interactions and that molecular crystals try to optimise their orientation with respect to the substrate lattice during their formation while retaining the optimum molecular bulk structure. [Dreher et al., Chem. Mater., 32, 20, 9034 (2020)]

While it might be true that single-molecule orientation on MoS₂ is still determined by the local corrugation of the M-S interaction potential, these studies show that it is negligible in comparison to the strength of intermolecular interactions for pentacene and perfluoropentacene [see also Félix et al., Cryst. Growth Des., 16, 12, 6941 (2016)]. Since the mutual attraction of the F₆PENs is stronger than the interactions between PEN or PFP molecules, the M-S interaction potential is even more negligible. Hence, we do not include it.

We have modified our discussion and explanation of the computational model to go into more detail for these arguments and hope that this satisfies the reviewers.

-Page 5, col1 and Supporting Information Sec 7, connected to previous point: a model is presented to find the optimal shape of the islands at fixed number of molecules (NoM). The optimal shape found contrasts the experimental results. With fixed NoM, the desorption is not considered. I wonder if this should, instead, be a parameter in the model, allowing the NoM to change.

Indeed, we calculated the optimal shape of islands based on their boundary free energies with a fixed NoM which, as the reviewer states, do not match the experimental results. This shows that no rearrangement of the molecules occurs. Hence, the island shape is kinetically controlled and can only be rationalised by modelling the desorption process as we do in the main paper. Former Sec 7 in the Supp. Inf. (now Sec 10) aims to support our assumption that no molecular rearrangement occurs even at elevated temperatures.

Moreover, have the Authors thought to perform annealing experiments at different temperatures, lower and higher than ~400K? In the work the M-S interaction is completely neglected, but a surface adsorption energy is present, thus it would be worth to check if annealing at different T could produce a molecular rearrangement (towards a deeper energy minimum), without desorption. A SE-extended shape could be found, better validating the claim of kinetic control.

We also have applied annealing temperatures at ~420K, which has led to a complete desorption of the film. For lower temperatures at ~380K the monolayer remains complete. By annealing only for a few seconds at ~400K, we have observed the very onset of desorption (see Fig. S12, Supp. Inf.). Thus, we conclude, that coverage control during desorption is best achieved by controlling annealing duration. However, it is quite time and material consuming to systematically control the island size.

These findings demonstrate that no rearrangement occurs. Instead, molecules will desorb once they are detached from the monolayer. This is in agreement with TPD measurements for the non-fluorinated and fully fluorinated pentacene [Kachel et al., Chem. Sci. 12, 2575 (2021)], where nominal monolayers of these compounds start to desorb at 400 K.

We have added STM data to the Supp. Inf. (Fig. S12) that shows the onset of desorption.

-The "References" section is divided in two parts, with the Methods section in between, this formatting should be corrected.

We thank the reviewer for this comment and have adapted the formatting.

-In the abstract/conclusions, the Authors describe theirs as a novel "approach." Although the work is interesting, it is not exactly clear that is meant by "approach." Moreover, recent studies - e. g. Commun Chem 1, 66 (2018) – analysed already in detail the possibility to use kinetics to increase the number of metastable self-assembled molecular network on substrates.

We thank the reviewer for this remark and for pointing out a study that we have not found during our literature research. In the referenced study, thermal treatment is used to achieve structural transitions in a covalently bound self-assembled monolayer on calcite. Thus, structural configurations are achieved that are not the energetic optimum. This is the only similarity to our work. In the referenced study, structure manipulation is

realised on a nanoscopic scale, as molecules either adopt a monomer or a dimer structure with covalent intermolecular and molecule-substrate bonds. In our study, we describe a mesoscopic structural control over the shape of nanosheets without changes in the molecular packing motif, i.e., the nanoscopic structure. For all observed nanosheet shapes, molecules adopt the same packing motif. To the best of our knowledge after an extensive literature research, this concept of mesoscopic structural control through kinetics of structure formation is new, hence our description as a 'novel approach' is justified.

However, it appears that the use of the term 'self-assembly' might lead readers to associate our work with what is commonly known as self-assembled monolayers, i.e., covalently bound molecular systems. To avoid this association, we have modified our title and avoided the term 'self-assembly' in the manuscript.

In summary, the manuscript is interesting, the quality of the analysis is overall valid, even if some strong approximations are adopted. The level of the presentation could be improved in some parts, e.g. the SE/LE terminology could be clearer. More explicit descriptions (e.g. in captions) would be beneficial. Sometimes there is a not an in-depth discussion on the results in the main text, especially concerning the electrostatic interactions at play, but this is complemented by the Supporting Information. Still, if the format allows, I would include in the main text some considerations present in the SI.

We thank the reviewer for this suggestion. However, since other reviewers suggest reducing the discussion of computational results in the main paper, we have not moved discussions from SI to the main paper.

In my opinion the results found, even if interesting, are specific to the field. The idea of using kinetics to obtain self-assemblies is not new and I am not sure if the sets of rules defined at the end could be effectively used for other systems, to control the shape of self-assembled islands (this is, in general, an extremely challenging task). The impact of the presented results makes the manuscript more suitable to other journals. For all these reasons, I do not recommend publication in Nature Communications. Alternative choices, after addressing all points suggested above, could be Communications Chemistry/Physics or Scientific Reports.

We firmly disagree with the reviewer's opinion on the relevance of our study. Firstly, the combination of TMDCs with van-der-Waals bound adlayers of organic semiconductors is an emerging and currently highly relevant field within the field of condensed matter physics and as such has had several focus sessions in recent renowned international conferences. There certainly is a large scientific interest in the topic.

In our revised manuscript, we extend our discussion and analysis of the mechanisms that permit shape control of 2D molecular islands that are physisorbed (van-der-Waals bound) to the substrate surfaces based on additional simulations for a partially fluorinated perylene, providing further insight into requirements for systems that allow such structural control. The fact that achieving such structural control is a challenging task does not diminish the relevance of our study, but proves that it represents an important advance in the development of strategies for structural control in low-dimensional systems.

Furthermore, as discussed in the second comment above, the novelty of our study lies not in the fact that we use thermal treatment to induce structural changes in molecular adlayers. As the reviewer has shown, this general concept has already been applied to achieve microstructural transitions in self-assembled monolayers. The novelty of our study is the possibility to change the mesoscopic shape of molecular aggregates that nonetheless have the same microstructure, i.e., molecular packing motif. This use of thermal treatment is fundamentally different from the study the reviewer cites and therefore by all means new.

For these reasons, we do believe that our study is well-suited for publication in Nature Communications, as it concerns a field of significant scientific interest and represents an important advance in that field.

Reviewer #3 (Remarks to the Author):

This work proposed a design strategy of controlling 2D organic monolayer on TDMC taking advantages of anisotropic Van de Waals interactions between functionalized acenes. STM was used to monitor the adsorption and desorption of nanosheets, and different shapes were observed accordingly. MD and MC simulations were performed for the attachment to and detachment from the nanosheet, respectively. And it was concluded that the desorption process is kinetic controlled because the free energy unfavorable LE extension shape is dominated. This is an interesting work, however the method used is not clarified and the analysis is confusing. I cannot recommend it for publication in its present form. Comments are given below.

1. Since the author claimed that MD results supported their conclusions, the details have to be provided for MD simulations, including the setup of simulation box, the algorithm, the ensemble used, etc. otherwise it is impossible to judge if the results obtained are reliable or not. Note that the MC simulations used for the detachment process is different from MD simulations.

We thank the reviewer for these remarks. The description of our simulations as MD simulations is mostly incorrect. Therefore, we have improved our terminology, focusing on the Monte Carlo approach. We have also extended the description of our algorithms and methods and uploaded our source code to an open access repository with a detailed documentation. We hope that the reviewer is now able to judge the reliability of our computational results.

2. When modeling the TDMC/OSC complex in simulations, the interaction between monolayer and the substrate was not included. Although for a single organic molecule, the intermolecular interactions could be much larger than the molecule/substrate interaction, this may not be true for larger aggregate. Also functionalized acene may have stronger interactions with the substrate than unfunctionalized one, and surface defects and/or reconstruction may also result in nonnegligible molecular substrate interactions. Systematic evaluation of the effect of molecule substrate interaction on the shape control is recommended.

We are grateful for this remark. Regarding the argument that molecule-substrate interactions might not be negligible for larger molecular aggregates, we refer to Ref. [Dreher et al., Chem. Mater., 32, 20, 9034 (2020)]. This study shows that crystallites of perfluoropentacene ($C_{22}F_{14}$) optimize intermolecular arrangement first. During formation of multilayers, the orientation of the bulk film is optimized with respect to the substrate while retaining the optimum molecular bulk structure. Therefore, the reviewer's argument is valid, but not relevant for our analysis and conclusions, as it shows that structure formation of molecular aggregates is not influenced by the substrate, only the orientation of the aggregates with respect to the substrate.

Regarding the argument of potentially different molecule-substrate interactions for differently functionalized acenes, we refer to Ref. [Kachel et al., Chem. Sci., 12, 2575 (2021)]. This study shows no significant difference between the molecule-substrate binding energies and surface dynamics of pentacene ($C_{22}H_{14}$) and perfluoropentacene. Of course, one might argue that the F_6 PENs studied here are again different molecules. However, due to the fact that optoelectronic properties of partially fluorinated acenes tend to lie in the range defined by non-fluorinated and perfluorinated species [Hofmann et al., Angew. Chem. Int. Ed., 59, 16501 (2020)], one can argue that molecule-substrate interactions should also be similar for our F_6 PENs. Moreover, the temperature window for activation of desorption from the first layers is not too different from that of pentacene and perfluoropentacene, indicating that the molecule-substrate binding energy is similar.

We do not want to argue that surface defects do not influence interactions with the substrate. However, they should not significantly influence structure formation of molecular aggregates, since they are spatially confined to single atoms and spaced sufficiently far apart (around 5-50nm, cf. Fig. S1 in SI) to render the occurrence of multiple defects within one of our small nanosheets unlikely. Since we are only interested in molecular arrangement, not in the orientation, position, and epitaxial relation of these aggregates on the substrates, we do not need to include molecule-substrate interactions in our model.

We have modified the description and discussion of our computational model to discuss the arguments above in greater detail.

3. It is argued that the sheet growth was governed by the nearest neighbor interaction, however, only the single molecule process was discussed. It is possible that the entire sheet breaks into several parts, and when the entropy contribution was included (the smaller sheet can rotate especially at elevated temperatures), the desorption process may actually be thermodynamic controlled. Indeed Figure 1e displays several sheets appearing in different orientations. Large scale STM images for the other functionalized molecule are helpful for further analysis.

We are aware that our model is greatly simplified by not considering entropy contributions or the breaking of islands or plenty of other effects. However, even in its simplicity, it can comprehensively reproduce experimental results on a qualitative level. An increase in complexity might increase accuracy and potentially allow some more quantitative conclusions, but not provide any added benefit regarding our central conclusions. Moreover, as shown in the main paper, the inner structure remains the same upon partial desorption. This indicates that no notable rearrangement processes occur.

I would also recommend taking the STM images at different times to monitor the adsorption/desorption processes (if possible).

While such measurements would certainly be interesting, they are unfortunately not possible. Firstly, the STM must operate at cryogenic temperature to enable molecular resolution of such weakly adsorbed adlayers, which is not compatible with thermally activated desorption and might completely suppress the formation of ordered molecular islands upon adsorption. In order to enable a more extensive experimental microscopic analysis of the ad- and desorption processes, it would require to examine the same spot on the sample under the microscope, which is hardly possible with intermediate heating steps due to thermal drift. Second, a deposition of molecules during STM measurements (at low temperatures) would not lead to the formation of ordered sheets but instead would result in disordered adlayers.

Reviewer #4 (Remarks to the Author):

The authors show that two-dimensional (2D) nanosheets of regioselectively fluorinated pentacene derivatives exhibit different shapes when formed by adsorption or desorption. They use molecular dynamics (MD) simulations based on an atomistic model and fundamental van der Waals (vdW) force fields to explain the kinetically controlled formation of 2D molecular nanosheets. They expand the approach to mesoscopic structural control of organic molecular nanosheets on molybdenum disulphide. I recommend to publish after the following questions to be addressed.

The STM micrographs of nominal monolayer and islands of L-F₆PEN had better exhibit in same range.

We agree that micrographs with the same range might be somewhat easier to interpret. However, as we discuss in more detail in the comment below, imaging of weakly bound organic islands at a scale that allows to image them completely with molecular resolution is highly challenging. Images of the quality of, for instance, Fig. 1e, where high resolution was achieved in a scan of approximately 150x150 nm, are therefore rare. If we want to show all STM data in the same range, we have to crop all larger STM micrographs to the range of the smallest one we show in the manuscript, thereby losing a lot of information from images where larger scan areas with high resolution were achieved. In our opinion, showing large scans such as Fig. 1e whenever possible is more important than having the same range for all images. Therefore, no changes to the range of STM micrographs were made.

Fig. 1d show much worse quality and should be updated.

We agree that the image quality of the deposited and unheated submonolayer is not the best. However, one cannot find many STM micrographs of such weakly interacting systems with good resolution and the large image size required to analyse island shapes. Imaging of these structures has proven to be rather challenging, at least with our setup that is limited to minimum temperatures of around 110K. One has to consider, that this study deals with submonolayers of molecules physisorbed on rather weakly interacting substrates. Especially for the directly deposited films (Figure 1d) lots of single, highly mobile ad molecules or small molecular cluster exist on the bare substrate areas, which dramatically decrease the tip stability. Resolving a large micrograph of a molecular island with molecular resolution is therefore highly challenging. Nevertheless, we managed to record images with molecular resolution for submonolayer coverages on a scale of more than 60nm. For the annealed film, we managed to obtain even larger micrographs, since single molecules, which are not attached to the islands anymore are desorbed already, hence the tip stability is much higher. Since we recorded such large micrographs we also wanted to demonstrate them to better confirm our results. Furthermore, since the molecules used in our study are not commercially available and their synthesis is challenging, their supply is limited and has depleted for these measurements.

Nevertheless, we managed to perform additional STM measurements of L-F₆PEN on MoS₂ for two different coverages (~10% and ~60% of a monolayer, see Fig. S11 in SI). The first shows similar $\langle \vec{b} \rangle$ - extended nanosheets as already discussed in the main paper. Because of fewer molecules on the surface, the STM was more stable and we managed to resolve an area of (120 x 120) nm (which is in a similar range as compared to Fig. 1e), such that the whole nanosheet and its surrounding is visible. Still, we were able to achieve molecular resolution within such a large scan area.

The scale bars of inner figures are missed.

We have added scale bars to all insets of STM images.

These islands of L-F₆PEN in Fig. 1d show different orientation, but they were in same orientation before annealed from the complete monolayer, the simulation of desorption kinetics in figure 3 not mention the phenomenon as well, could the authors give some explains?

We would like to emphasise that due to the 6-fold symmetry of the MoS₂ surface the identified $\begin{pmatrix} 1.64 & 1.00 \\ 5.76 & -4.50 \end{pmatrix}$ point-on-line superstructure of the L-F₆PEN adlayer occurs in rotational as well as mirror domains. Therefore, the domains exhibit characteristic angles with respect to the substrate azimuth direction. Since after the heating steps we cannot re-approach the STM tip to the same position, different regions are imaged. As for the directly grown islands, we image different regions to make sure that our analysis is representative, but observe different rotational domains. Regarding the islands shown in Fig. 1e (partially desorbed L-F₆PEN monolayer), we believe that the different molecular orientations in different islands were already present in the initial monolayer as different rotational domains. One evidence for this is the fact that neighbouring islands, for instance the three long 'vertical' ones in the centre of the image or the two bottom islands at the left edge of the image, show equal molecular orientations.

2. As showed in Fig. 1d, submonolayer films with a nominal thickness of 0.8Å form well-ordered islands. The authors had better offer some statistical data of these islands, containing the area and aspect ratio. The STM micrographs and statistical data of different nominal thickness should also be showed.

We agree with the reviewer, that more insights into the molecular island growth could be gained by systematically analysis of different film thicknesses. It is, however, quite challenging to image a whole molecular island (to determine its aspect ratio) with high molecular resolution. Especially the directly adsorbed films exhibit single, diffusive molecules which are highly mobile and are likely to harm the tip quality (as one can see for 0.8Å L-F₆PEN in Fig. 1d). Therefore, we cannot provide a whole series of high quality STM images at 10 or more film thicknesses. However, we performed STM measurements of films with two further surface coverages in Figure S11 (*Supp. Inf.*) to show the island growth at very low and high coverages of L-F₆PEN on MoS₂. At 0.4Å nominal thickness, similar to 0.8Å, isolated islands are formed which are elongated along $\langle \vec{b} \rangle$. Interestingly, these islands are separated of more than 50nm. So it appears, that only the density of islands on the surface is affected by the smaller coverage. At high coverages around 2.0Å the islands start to merge together already.

3. Fig. 3b shows a map of the average desorption sequence of molecules from an L-F₆PEN nanosheet of mLE=20 by nSE=50 molecules. Why did the authors choose 20*50 nanosheet? What is typical shape and aspect ratio of complete monolayer?

We have chosen this specific nanosheet shape because it is basically square. Thus, we avoid a bias by starting with a nanosheet that is extended in the direction of a unit vector. Thus, our results better show that a preferential shape is formed due to desorption. We have also tested other initial shapes, as discussed in the *Supp. Inf.* and find the same results.

The monolayer domains are commonly extended over more than 100nm. Therefore, it is hardly possible to resolve a whole monolayer domain including its complete boundary. However, within such a large monolayer, molecules start to desorb also from within the ordered molecular structure and favour desorption from such a single vacancy in direction $\langle \vec{b} \rangle$. We have added a STM image in Figure S9 in SI, which shows the beginning desorption process.

4. The authors claim that their novel approach to structural control in 2D molecular nanosystems can be applied to a vast number of molecular materials on weakly interacting substrates. Could they offer some examples in simulation or experiment?

Stimulated by the question of the reviewers to base our main conclusion and our formulated design rules on a somewhat broad data situation, we examined yet another molecular system. In addition to the two experimentally examined partially fluorinated pentacene derivatives, we have extended our computational analysis also to a partially fluorinated perylene (1,2,3,10,11,12-hexafluoroperylene (L-F₆PER)). This molecule does not exhibit such a strong geometrical anisotropy as compared to the L- and M-F₆PEN but is shaped almost square. Since this molecule has not been synthesised yet, we cannot provide experimental data. Computational structure optimisation suggests two stable molecular packing motifs for L-F₆PER. Qualitatively, these packing motifs are similar to the unit cells of the L- and M-F₆PEN, one non- and one alternating alignment. Statistical analysis of the adsorption process of F₆PER shows a similar behaviour as the F₆PENs. For the non-alternating unit cell an attachment at the *LEs* is favoured (as in the case of the L-F₆PEN). In contrast, the alternating unit cell favours attachment at the *SEs* (as in the case of the M-F₆PEN). This demonstrates that not only the geometrical anisotropy of the molecule plays a role for the island shape, but also the 2D crystal structure. Thus, it is evident that not only the electrostatic anisotropy of the molecule itself, but also the molecular packing motif are important for the kinetics of structure formation.

Reviewers' Comments:

Reviewer #1:

Remarks to the Author:

I have thoroughly studied the revised manuscript and the authors' response to the reviewers' comments. The authors presented robust and convincing arguments and fully and satisfactorily responded to all concerns raised by the reviewers. The revised manuscript can be accepted for publication in its current form.

Reviewer #2:

Remarks to the Author:

In the resubmitted version of the manuscript, the Authors, among other things, have improved the clarity and readability of the main text, which represented a major issue in the original version. The language in some parts can be still amended, but only with minor changes (please see below). They also extended the computational analysis to a new molecule, denoted L-F6PER (not synthesized, and no experiments were done on it).

However, they still do not include in their model any molecule-substrate (M-S) interaction, giving several arguments against. Considering that the Authors enjoy the "maximum-flexibility" (see reply to Reviewer 1) of their in-house programmed code, it is surprising that, as a more direct proof of their conclusions, they don't consider this inclusion. A more refined model would represent a direct proof to validate their work, rather than relying only on (maybe sound) considerations or on previous literature.

Please see a list of points below (not in order of importance):

-Page 3, col 2: The following sentences should be amended to further improve clarity:

"...molecules located at the LEs of a nanosheet have the weakest bond to the nanosheet. This is caused by the strong mutual attraction of molecules along their long sides. While molecules located at the SEs have two long-side nearest neighbours, those at the LEs have only one long-side neighbour and are therefore significantly more likely to desorb than molecules located at the SEs."

→ "...molecules located along the LEs of a nanosheet have the weakest bond with the nanosheet.

While molecules located along the SEs have two long-side nearest neighbours (with a stronger mutual attraction between molecules) those along the LEs have only one long-side neighbour and are therefore significantly more likely to desorb than molecules located along the SEs"

-Page 4, col 1: "Here, molecular islands are preferentially ...(cf. Fig. 3c)"

→ "Here, as for growth/adsorption, molecular islands are preferentially(cf. Fig. 3c), unlike in the L-F6PEN case."

-Page 4, col 2: "...there are weaker but longer ranged attractive interactions along the SEs." Probably the longer range is due to dipolar nature of the interaction along the  direction, unlike the inherently quadrupolar interaction along the direction? The Authors could comment on that in the text.

-Page 4, col 2: "The reason... are almost equal at all edges." A comment: by applying the very same logic that the Authors used for the L-F6PEN nanosheets, it seems that the M-F6PEN molecules located along the LEs have four "points" of electrostatic attraction, while the ones located along the SEs they have five. However, this might have a small effect due to the reduced strength of the attractions. The Authors could comment on that in the text.

-Page 5, col 2: The very same notation should be kept throughout the entire text, for clarity, e.g.

"Along the uniformly terminated edges...molecules. At the same time, molecules at the uniformly terminated edges have, at first glance somewhat paradoxically, the weakest bond to the nanosheet and therefore tend to desorb first, leading to the formation of islands that are elongated in direction of the unit vector parallel to these edges."

→

"Along the LE edges....molecules. In this case the molecules, at first glance somewhat paradoxically (but analogously to detachments from L-F6PEN nanosheets), have the weakest bond to the nanosheet and therefore tend to desorb first, leading to the formation of islands elongated in the  direction."

-Whenever referring to the Supp. Inf. , please indicate the Section/Page of the relevant information.

-Page 6, col 1: "Since the mechanism of nanosheet shape control identified here is essentially based on the electrostatic part of the intermolecular van der Waals interactions, it is hardly applicable to molecular films adsorbed on metal substrates, where additional molecule substrate interactions occur that affect the molecular thin film structure."

and from their rebuttal

"The term 'vdW interactions' refers to a range of interactions that includes electrostatic interactions, Pauli repulsion and dispersion forces" .

Two comments: 1. The Authors suggest the issue of using metal substrates where the molecule-substrate interactions are strong. I would add, for metals, that the electronic screening could, in principle, wash out the anisotropies of the Coulomb electrostatic interactions, making the methodology unfeasible.

2. the vdW interaction is different from the electrostatic interaction. This has to be corrected (everywhere in the work) as the statement quoted above is wrong. The forces dominating the formation of nanosheets in the manuscript are predominantly of electrostatic nature, not of vdW nature.

From their rebuttal:

"we would first like to point out that the state shown in Fig. 1d is closer to the energy local minimum (with a short LE edge) than the state in Fig. 1e. " Authors should justify this statement (also) in the main text.

In summary, I find that the manuscript has improved, and the message is conveyed more clearly. The Authors clarified also why their approach would be of general interest, rather than confined to a specific field. One aspect seems still critical, where only indirect arguments were provided: completely neglecting M-S interactions and their effects. There seems to be is no smoking gun proof that they can be neglected. For all these reasons, I would recommend publication in Nature Communications, if all points above are addressed and if M-S if effects are included in some form in their computational modelling.

Reviewer #4:

Remarks to the Author:

The authors have addressed the questions and revised the manuscript appropriately. Therefore, the revised manuscript is acceptable for publication.

REVIEWERS' COMMENTS

Reviewer #2 (Remarks to the Author):

In the resubmitted version of the manuscript, the Authors, among other things, have improved the clarity and readability of the main text, which represented a major issue in the original version. The language in some parts can be still amended, but only with minor changes (please see below). They also extended the computational analysis to a new molecule, denoted L-F6PER (not synthesized, and no experiments were done on it).

However, they still do not include in their model any molecule-substrate (M-S) interaction, giving several arguments against. Considering that the Authors enjoy the "maximum-flexibility" (see reply to Reviewer 1) of their in-house programmed code, it is surprising that, as a more direct proof of their conclusions, they don't consider this inclusion. A more refined model would represent a direct proof to validate their work, rather than relying only on (maybe sound) considerations or on previous literature.

Please see a list of points below (not in order of importance):

-Page 3, col 2: The following sentences should be amended to further improve clarity:

"...molecules located at the LEs of a nanosheet have the weakest bond to the nanosheet. This is caused by the strong mutual attraction of molecules along their long sides. While molecules located at the SEs have two long-side nearest neighbours, those at the LEs have only one long-side neighbour and are therefore significantly more likely to desorb than molecules located at the SEs."

→ "...molecules located along the LEs of a nanosheet have the weakest bond with the nanosheet. While molecules located along the SEs have two long-side nearest neighbours (with a stronger mutual attraction between molecules) those along the LEs have only one long-side neighbour and are therefore significantly more likely to desorb than molecules located along the SEs"

We thank you for the constructive comments and have amended the relevant text in the manuscript accordingly.

-Page 4, col 1: "Here, molecular islands are preferentially ...(cf. Fig. 3c)"

→ "Here, as for growth/adsorption, molecular islands are preferentially ...(cf. Fig. 3c), unlike in the L-F6PEN case."

We thank you for the constructive comments and have amended the relevant text in the manuscript accordingly.

-Page 4, col 2: "...there are weaker but longer ranged attractive interactions along the SEs." Probably the longer range is due to dipolar nature of the interaction along the  direction, unlike the inherently quadrupolar interaction along the direction? The Authors could comment on that in the text.

We thank the reviewer for the comment. Indeed, the bipolar electrostatic potential of the M-F6PEN molecule can lead to different macroscopic potentials in the film, whether the molecules are uniformly or alternating aligned. In this particular case of M-F₆PEN the potential is alternating along the LE direction. Therefore, in a simple picture, the small cusps of alternating attractive and repulsive interaction at the LE will cancel out in the far field, but not in the near field. However, we would not call it "quadrupolar" or "dipolar", since this the potential landscape is much more complicated to simplify it as a pure dipole or quadrupole expansion. Nevertheless, considerations about the interactions in the near and far field can help to understand the Monte Carlo simulations. We comment on that after the next reviewer point as well as in the manuscript.

-Page 4, col 2: "The reason... are almost equal at all edges." A comment: by applying the very same logic that the Authors used for the L-F₆PEN nanosheets, it seems that the M-F₆PEN molecules located along the LEs have four "points" of electrostatic attraction, while the ones located along the SEs they have five. However, this might have a small effect due to the reduced strength of the attractions. The Authors could comment on that in the text.

We guess the reviewer is addressing the attachment energy map (AEM) of an M-F₆PEN island in Figure 3d. Counting the blue (attractive) local minima in energy at the island boundary, there are five short-ranged minima along the LE and four long-ranged minima along the SE. Simply counting the number of local minima, one would expect an adsorption at the LE, opposite to the experiment. However, the spatial range of high attraction seems to be more relevant here. At small surface coverages, molecules are unlikely to adsorb on the surface in close proximity to an island edge during deposition. Instead, they have to diffuse from their adsorption site towards an island to attach to it, probing only the longest-range interaction potential of the island. The strong but spatially confined energetic minima of these potentials are of comparatively short range due to the directly neighbouring repulsive areas of the potential. Therefore, molecules are statistically most likely to attach to the edges of an island with the longest range of the interaction potential, which in this case are not the absolute minima of the complete potential map, but only local minima, as demonstrated by our Monte Carlo simulations of the single-molecule attachment process.

This consideration also explains differences in the homogeneity of the attachment probability maps of L-F₆PEN and M-F₆PEN (Figs. 2c and 3e). For L-F₆PEN, the relative size of the repulsive areas of the potential as compared to the attractive areas increases with increasing b extension. This leads to a dependency of the attachment probabilities on the island shape. By contrast, for M-F₆PEN, the relative size of repulsive over attractive areas does not change. Hence, the attachment probability map is more homogeneous.

We have extended our discussion in the manuscript.

-Page 5, col 2: The very same notation should be kept throughout the entire text, for clarity, e.g. "Along the uniformly terminated edges...molecules. At the same time, molecules at the uniformly terminated edges have, at first glance somewhat paradoxically, the weakest bond to the nanosheet and therefore tend to desorb first, leading to the formation of islands that are elongated in direction of the unit vector parallel to these edges."

→

"Along the LE edges....molecules. In this case the molecules, at first glance somewhat paradoxically (but analogously to detachments from L-F₆PEN nanosheets), have the weakest bond to the nanosheet and therefore tend to desorb first, leading to the formation of islands elongated in the a direction."

We amended the regarding sentence in the manuscript.

-Whenever referring to the Supp. Inf., please indicate the Section/Page of the relevant information.

For all references to the Supp. Inf., we have added the exact section.

-Page 6, col 1: "Since the mechanism of nanosheet shape control identified here is essentially based on the electrostatic part of the intermolecular van der Waals interactions, it is hardly applicable to molecular films adsorbed on metal substrates, where additional molecule substrate interactions occur that affect the molecular thin film structure." and from their rebuttal: "The term 'vdW interactions' refers to a range of interactions that includes electrostatic interactions, Pauli repulsion and dispersion forces".

Two comments: 1. The Authors suggest the issue of using metal substrates where the molecule-substrate interactions are strong. I would add, for metals, that the electronic screening could, in principle, wash out the anisotropies of the Coulomb electrostatic interactions, making the methodology unfeasible.

We agree with the reviewer that on metals an additional electronic screening could occur, making the description even more difficult. However, it is not simply given that the presence of such an electronic screening washes out the anisotropies of the Coulomb electrostatic interactions completely. There are numerous examples, such as PTCDA/Au(111) [Kröger et al., J. Chem. Phys, 135, 234703 (2011)], or PFP+CuPc blend on Ag(111) [Goiri et al., Phys. Rev. Lett., 112, 117602 (2014)], which demonstrate that (electrostatic) intermolecular interactions are a key factor for the resulting structure pattern within a monolayer.

2. the vdW interaction is different from the electrostatic interaction. This has to be corrected (everywhere in the work) as the statement quoted above is wrong. The forces dominating the formation of nanosheets in the manuscript are predominantly of electrostatic nature, not of vdW nature.

In general, the reviewer is right that vdW and electrostatic interaction are different. However, we would like to note that vdW interaction contains not induced-dipole interactions such as London dispersion forces (fluctuating dipole – induced dipole) and Debye forces (permanent dipole – induced dipole), but also interactions between permanent dipoles (so called Keesom forces) as introduced by W.H. Keesom (see Phys. Z. 22, 129-141 (1921), Phys. Z. 22, 643-644 (1921), Phys. Z. 23, 225-228 (1922)). Therefore, we disagree with the classification made by the reviewer, but admit that our wording was somewhat imprecise since we do not consider Coulomb interactions between charged molecules here. To clarify this aspect we changed the wording “electrostatic part” by “Keesom forces”.

From their rebuttal:

“we would first like to point out that the state shown in Fig. 1d is closer to the energy local minimum (with a short LE edge) than the state in Fig. 1e. ” Authors should justify this statement (also) in the main text.

We thank the reviewer for the comment and added such a statement in the main text: “Considerations about the boundary free energy (the 2D analogue of the surface free energy, cf. Supplementary Note 5) suggest that the $\langle \vec{b} \rangle$ -extended sheets are much closer to the energy local minimum. Hence, it appears counterintuitive, that $\langle \vec{b} \rangle$ -extended sheets are found experimentally prior, but not after thermal treatment.” (p. 3, top of left column).

In summary, I find that the manuscript has improved, and the message is conveyed more clearly. The Authors clarified also why their approach would be of general interest, rather than confined to a specific field. One aspect seems still critical, where only indirect arguments were provided: completely neglecting M-S interactions and their effects. There seems to be is no smoking gun proof that they can be neglected. For all these reasons, I would recommend publication in Nature Communications, if all points above are addressed and if M-S if effects are included in some form in their computational modelling.

We would like to remind that the strength of the molecule-substrate interaction is only relevant for the activation energy of desorption. While on metals or other more reactive substrates, where the molecular packing in monolayer films is indeed affected by the molecule-substrate interaction this effect can be excluded on the weakly interacting substrates like the presently studied MoS₂. Furthermore, since the lateral corrugation of the M-S interaction is very small, which has been shown both theoretically for PEN/MoS₂ and experimentally for PFP/MoS₂, the packing motif and island growth is only determined by the anisotropy of the lateral M-M interaction. In this respect, the exact strength of the M-S interaction plays no role for the phenomena discussed here. Due to the lateral extent of acenes, one molecule interacts with several substrate atoms simultaneously, making the effective corrugation of the M-S interaction much smaller than for small molecules (such as CO) or SAMs that are covalently bonded to the substrate through one atom.